# Mice with a diverse human T cell receptor repertoire selected on multiple HLA class I molecules

Arunraj Dhamodaran [1,2,4], Xiaojing Chen[1], Niklas Fellmer[2], Deepti Agrawal[2], Eric Danner[2], Ralf Kühn [3] & Thomas Blankenstein [1] ✉

T cell receptor (TCR) gene therapy is an effective cancer treatment. Ideally, the TCR should be of human origin and have optimal avidity, e.g., isolated from a tumor antigen-non-tolerant host. Previously, we developed AB*ab*-A2 mice which carry human TCRα and TCRβ gene loci and the human leukocyte antigen class I gene HLA-*A*02:01* and are deficient for the corresponding mouse genes. Into these mice, we here introduce by PiggyBac transposon HLA-*A*03:01*, -*A*11:01*, -*B*07:02*, -*B*15:01*, -*C*04:01*, and -*C*07:02* genes. These mice, termed AB*ab*-I, exhibit increased peripheral CD8⁺ T cell counts and a higher CD8/CD4 ratio compared to AB*ab*-A2 mice. AB*ab*-I mice display a broader TCR repertoire with more unique V(D)J-TCRß clonotypes than AB*ab*-A2 mice. Multi-HLA-I expression selected, on average, TCR with longer complementary determining region 3 (CDR3) compared to expression of a single HLA-I. AB*ab*-I mice mount robust immune responses against viral, tumor-associated, and tumor-specific antigens. AB*ab*-I mice allow simultaneous epitope and TCR discovery with broad HLA coverage, which could increase the number of cancer patients amenable to TCR-T treatments.

The adoptive T cell therapy (ATT) with engineered T cell receptor (TCR) has great potential to treat cancer[1]. TCR-modified CD8⁺ T cells can provide high efficacy compared to other cancer treatments when directed toward tumor-specific antigen (TSA) or tumor-associated antigen (TAA) presented on HLA-I molecules[2–4]. TSA-specific HLA-I TCRs can be isolated from human donors or transgenic mice[5–9]. TAA-specific HLA-I TCRs can preferentially be isolated from transgenic mice because humans are typically tolerant to TAAs, which are often self-peptides. During thymic selection, high-avidity T cells recognizing self-peptides are deleted to prevent autoimmunity, leaving only low-avidity T cells in circulation. These low-avidity T cells are less effective in mounting a strong immune response against tumors, limiting their therapeutic potential. Transgenic mice are suitable for isolating TCRs from a naïve repertoire, which, upon immunization, becomes

activated and differentiates into memory T cells, from which TCRs can ultimately be isolated[10–12]. Chimeric HLA-I transgenic mouse models existed for some time, but questions remain concerning inefficient selection of mouse TCRs by human HLA molecules, overall TCR repertoire size as a result of partial mismatch between mouse and human HLA-TCR molecules, and loss of certain Vαß specificities during thymic selection[13–15]. Besides, murine TCRs from HLA-transgenic mice could be immunogenic when transferred to humans[16]. To enable full human TCR discovery from transgenic mice, we previously created a mouse model (AB*ab*-A2 mice, originally named AB*ab*DII) that bears the complete human *TCRα* and *TCRβ* gene loci and HLA-*A2* as a single HLA-I allele[7]. These mice were additionally deficient for mouse *tcra* and *tcrβ* as well as mouse MHC-I expression (*H2-D*ᵇ and β2-microglobulin (*ß2m*) double knockout). TCR gene loci humanization in AB*ab*-A2 mice

[1]Molecular Immunology and Gene Therapy, Max Delbrück Center for Molecular Medicine in the Helmholtz Association (MDC), Berlin, Germany. [2]T-knife Therapeutics, Inc., San Francisco, USA. [3]Genome Engineering and Disease Models, Max Delbrück Center for Molecular Medicine in the Helmholtz Association (MDC), Berlin, Germany. [4]Present address: Microbiome and Cancer Division, German Cancer Research Center in the Helmholtz Association (DKFZ), Heidelberg, Germany. ✉e-mail: tblanke@mdc-berlin.de

yielded functional CD8[+] T cell populations for optimal affinity TCR isolation, but the mice were limited to an HLA-A2 restricted repertoire.

To broaden the HLA-I repertoire and to study the effect of multiple HLA alleles on TCR selection, we here further engineered AB*ab*-A2 mice with six additional human HLA-I alleles by inserting them as one genotype. We achieved this by using PiggyBac transposon technology to insert a distinct set of six alleles into transcriptionally active sites. We designed the transgene construct containing HLA-*A\*03:01*, *A\*11:01*, *B\*07:02*, *B\*15:01*, *C\*04:01*, and *C\*07:02* with each HLA being a chimeric human-mouse fusion monochain containing α1 and α2 heavy chain cDNA regions from respective human HLA alleles, while α3, transmembrane and cytoplasmic domains from murine *H-2 D^b* gene and fused to the human *β2m* gene.

The novel AB*ab*-I mice exhibit increased thymic selection and peripheral counts of CD8[+] T cells and efficiently present epitopes ex vivo on all six HLA haplotype proteins. AB*ab*-I mice mount robust CD8[+] T cell responses in vivo against viral (Cytomegalovirus (CMV), Human Papillomavirus (HPV)), tumor-associated antigens (MAGE-A12), and tumor-specific neoantigens (mutant KRAS and CALR). This study establishes the AB*ab*-I mouse model as a high-throughput platform for simultaneous epitope and TCR discovery against epitopes presented by diverse HLAs in an unbiased fashion, thereby allowing, in the future, cancer patients beyond those expressing HLA-A2 to benefit from ATT treatments. AB*ab*-I mice are also useful for investigating human T cell biology and thymic selection under conditions of one versus multiple different HLA-I molecules.

## Results

### Generation of mice transgenic for human HLA class I haplotype

To broaden the HLA repertoire of humanized TCR gene loci mice[7], PiggyBac (PB) transposon technology was used to introduce the six HLA class I alleles HLA-*A\*03:01*, -*A\*11:01*, -*B\*07:02*, -*B\*15:01*, -*C\*04:01*, and -*C\*07:02* on a single gene construct to generate 'AB*ab*-I' mice (Fig. 1). These HLA alleles were chosen due to their high frequency

among the world population and because in silico predictions using the NetMHC algorithm[17] suggested that they are relatively frequent binders of potential recurrent neoantigens. This could be tested in these mice and potentially allow the isolation of neoantigen-specific TCRs. The six HLA genes were cloned in tandem as cDNA, each containing 5′ the H-2 promoter and 3′ the bovine growth hormone polyadenylation site. Each HLA gene contained the alpha-3 domain of mouse *H2-D^b* to ensure binding to mouse CD8α on T cells and was fused by a linker to the human *β2m* gene. This six-HLA gene cassette was cloned into the PB vector with inverted tandem repeat (ITR) at either site (Fig. 1). The PB ITR-flanked targeting vector carrying six HLA genes was co-injected with the hyperactive PB transposase mRNA into fertilized AB*ab*-A2 mouse oocytes. The PB transposase acts on the ITR-flanked targeting vector (cassette-bearing HLAs) via a cut-and-paste mechanism. The transposase enzyme, delivered as mRNA, excises the 5′ and 3′ ITR regions to release the HLA insert, which is then randomly integrated into transcriptionally active sites of the mouse genome (Fig. 1). After pronuclei injection, transgenic mice were generated and allele-specific PCRs confirmed the presence of the six HLA alleles in the founder animals (Supplementary Fig. 1). The resulting mice contained the human TCR gene loci transmitted from the AB*ab*-A2 genetic background as well as deficiency of mouse *tcraβ* and mouse MHC-I expression.

### Improved T cell development in AB*ab*-I compared to AB*ab*-A2 mice

To investigate the effect of multiple HLA alleles on human TCR T cell development, we analyzed thymic subsets of AB*ab*-I mice compared to single-HLA-bearing AB*ab*-A2 and wild-type C57BL6/N control mice. AB*ab*-I and AB*ab*-A2 mice had similar levels of CD4[+]CD8[+] double-positive thymocytes compared to control mice (Fig. 2A, B and Supplementary Fig. 2A). AB*ab*-I mice contained more CD8[+] single-positive cells ($6.3 \pm 2.27 \times 10^4$) than AB*ab*-A2 mice ($1.3 \pm 0.8 \times 10^4$), revealing that multiple HLA alleles selected large numbers of non-HLA-A2

PiggyBac transgene cassette (transposon size: 15,084 bp)

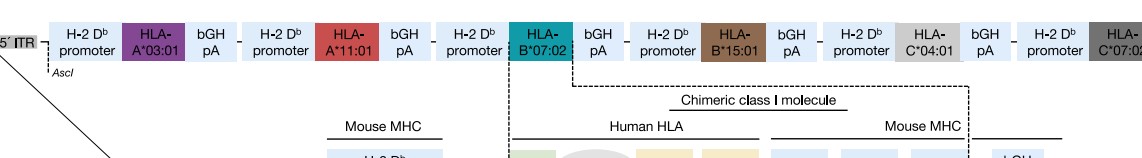

**Fig. 1 | Generation of AB*ab*-I mice by PiggyBac transposon system.** Schematic representation of PiggyBac (PB) transposon strategy for the generation of AB*ab*-I mice using pronuclei microinjection technology. The PB targeting AB*ab*-I transgene (flanked by ITRs for PB transposase catalysis) has repeating elements of six HLA ORFs constructed one after another (5′-to-3′) each with its own promoter and 3′-UTR components. Each HLA ORF is a human-mouse fusion monochain containing α1 and α2 heavy chain cDNA regions from respective human HLA alleles, while α3, transmembrane (TM), and cytoplasmic domain (CYT) from murine *H-2 D^b* gene fused by a glycine-serine linker to human β2 microglobulin (*β2M*). Created in BioRender. Dhamodaran, A. (2025) https://BioRender.com/wii3mup.

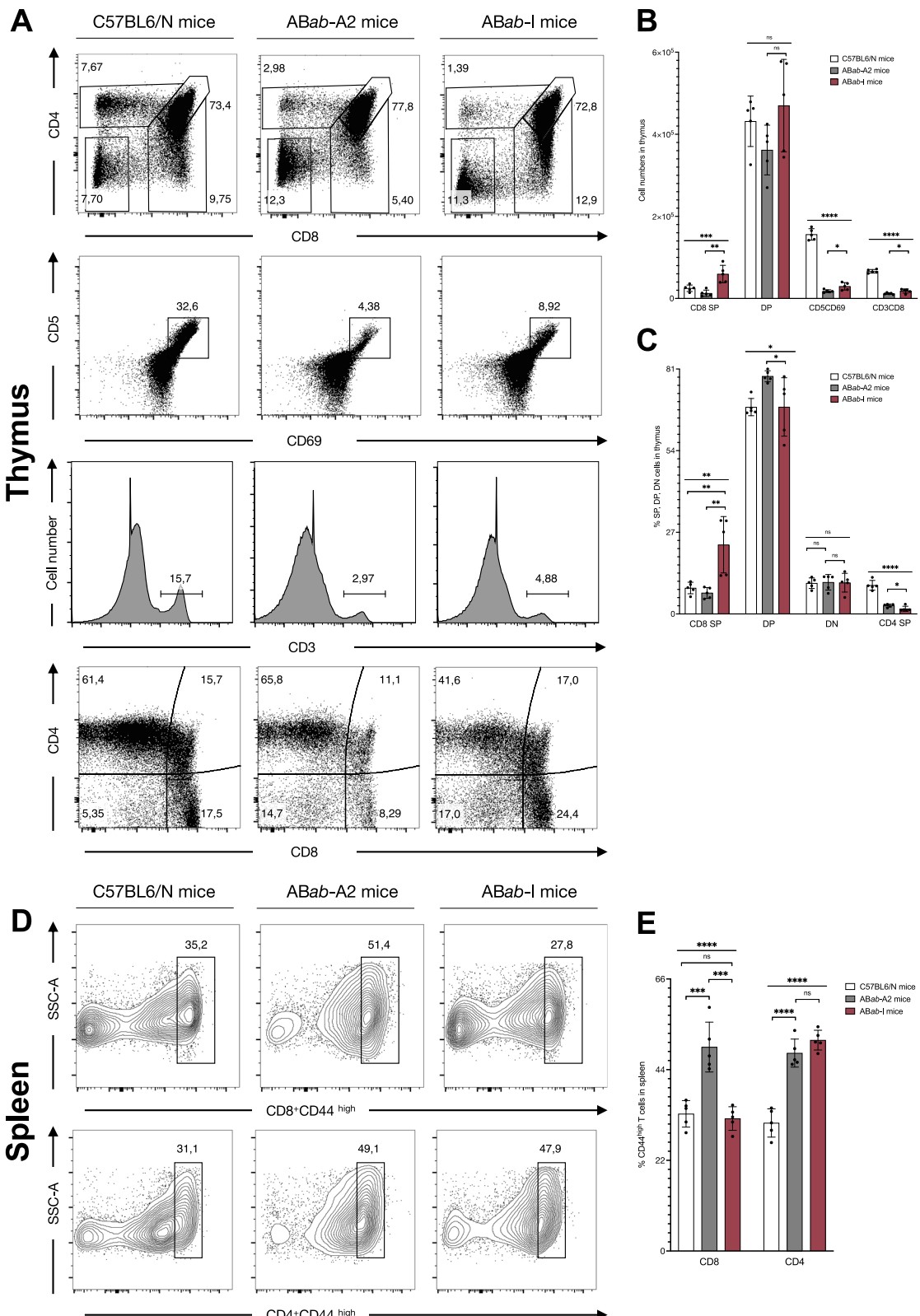

restricted thymocytes, which likely could not have been selected in AB*ab*-A2 mice (Fig. 2A, B). AB*ab*-I mice contained more CD5⁺CD69⁺ thymocytes ($3.1 \pm 0.9 \times 10^4$) than AB*ab*-A2 mice ($1.8 \pm 0.3 \times 10^4$), indicating that more T cells acquired a signal during positive and negative selection (Fig. 2A, B and Supplementary Fig. 2B). On average, 68% of double-positive thymocytes progressed to become 23% CD8⁺ single-

positive cells in AB*ab*-I mice, whereas only 7% cleared thymic selection in AB*ab*-A2 mice from 79% double-positive thymocytes (Fig. 2C). Furthermore, AB*ab*-I mice had overall improved T cell development with significant elevation in CD3⁺/CD8⁺ T cell numbers compared to AB*ab*-A2 mice (Fig. 2 and Supplementary Fig. 2A–D). From the periphery, we quantified the percentages of CD44^high splenocytes in AB*ab*-I versus

**Fig. 2 | T cell development in AB*ab*-I, AB*ab*-A2, and C57BL6/N mice.**
**A** Representative flow cytometry analysis of thymocyte staining from AB*ab*-I, AB*ab*-A2, and C57BL6/N mice (*n* = 5). 1st lane: CD4⁺ and CD8⁺ single- and double-positive cells, gated on total thymocytes; 2nd lane: cell percentages of CD5⁺CD69⁺ cells, gated on lymphocytes; 3rd lane: percentages of CD3⁺ cells; 4th lane: CD4⁺ and CD8⁺ single- and double-positive cells gated on CD3⁺ cells. **B** Quantified cell numbers of thymocyte populations: CD8SP, DP, CD5CD69, and CD3CD8 in respective mouse strains. **C** Quantified percentages of thymocyte populations: CD8SP, DP, DN, and CD4SP in respective mouse strains. **D** Representative flow cytometry analysis of splenocyte staining from AB*ab*-I, AB*ab*-A2, and C57BL6/N mice (*n* = 5), gated on CD3⁺ lymphocytes. 1st lane: CD8⁺CD44^high cells; 2nd lane: CD4⁺CD44^high cells. **E** Quantified percentages of CD44^high CD4⁺ and CD8⁺ populations from AB*ab*-I, AB*ab*-A2, and C57BL6/N mice (*n* = 5). Data in (**B**), (**C**), and (**E**) are presented as mean ± SD. Multiple group statistical comparisons were performed by one-way ANOVA, followed by pairwise comparisons using two-tailed unpaired Student's *t*-test. *P* values indicate: *$P < 0.05$; **$P < 0.01$; ***$P < 0.001$; ****$P < 0.0001$; ns not significant. Exact *P* values and source data are available in the Source Data file.

AB*ab*-A2 mice in comparison to control mice. CD44 is a marker for homeostatic proliferation. Higher CD44^high T cell numbers in mice represent lymphopenic conditions[18].

In splenocytes, we observed no significant change in CD44 expression among CD4⁺ T cells in both AB*ab*-I and AB*ab*-A2 mice (Fig. 2D, E and Supplementary Fig. 2E). However, CD44 expression by CD4⁺ T cells of both mouse lines was increased in comparison to control mice, which is likely caused by a partial mismatch between mouse and human TCR-MHC interaction and impaired CD4⁺ T cell development, since the human TCRs in AB*ab*-A2 and AB*ab*-I mice are selected by mouse MHC-II molecules[15]. AB*ab*-I mice had similar percentages of CD44^high CD8⁺ T cells as control mice, resembling a natural unbiased T cell development in the CD8⁺ T cell compartment (Fig. 2E). However, the substantial reduction of CD44 expression on CD8⁺ T cells in AB*ab*-I mice (32 ± 2.8%) compared to AB*ab*-A2 mice (49 ± 6%) showed that positive thymic selection by multiple HLA-I molecules was more effective and rescued from the lymphopenic condition seen when there is only a single HLA-I allele (Fig. 2D, E and Supplementary Fig. 2F).

### HLA haplotype expression in AB*ab*-I mice

HLA constructs, designed as single monochains in the same configuration and under the same H-2 promoter as in AB*ab*-I mice, were transfected into MCA205 cells and analyzed for HLA expression using anti-human pan HLA-ABC and ß2m antibodies. These data showed that each of the six HLA-I genes was well expressed on mouse cells (Supplementary Figs. 3A and 4A). The presence of interferon-stimulated response element (ISRE) in the H-2 promoter facilitated the upregulation of HLA-I expression upon IFN-γ treatment (Supplementary Fig. 3A). Staining with anti-human ß2m antibody resulted in a slightly elevated mean fluorescence intensity (MFI) range compared to the pan HLA-ABC antibody (Supplementary Fig. 3B). Due to its specificity for the amino terminus of β2m, the pan HLA-ABC antibody poorly stained recombinantly expressed human β2m molecules. Next, HLA surface proteins were stained and quantified in the peripheral blood cells of C57BL6/N, AB*ab*-A2, and AB*ab*-I mice using anti-human ß2m antibody (Fig. 3A and Supplementary Fig. 4B). 7.27% of lymphocytes from AB*ab*-A2 mice, but 40.1% of lymphocytes from AB*ab*-I mice stained positive for HLA expression using the ß2m antibody that targets all HLA-I molecules (Fig. 3A). MFI comparison of ß2m revealed 3 times more HLA expression in AB*ab*-I compared to that in AB*ab*-A2 mice (Fig. 3B, C). These ß2m staining data suggest HLA expression on the surface of AB*ab*-I lymphocytes.

### High CD8⁺ T cell numbers in AB*ab*-I mice

Peripheral T cells were analyzed between AB*ab*-I, AB*ab*-A2 and control mice (Fig. 4). Within CD3⁺ peripheral blood cells, there were 13% CD8⁺ T cells in AB*ab*-A2, 45% in AB*ab*-I and 41% in C57BL6/N control mice (Fig. 4A and Supplementary Fig. 4C). CD4⁺ T cells were 63% in AB*ab*-A2, 41% in AB*ab*-I and 53% in control mice (Fig. 4A). A significant number of AB*ab*-I mice accumulated 4-times more CD8⁺ T cell numbers in the periphery than AB*ab*-A2 mice (Fig. 4B). No significant difference in the CD4⁺ T cell numbers was observed between AB*ab*-A2 and AB*ab*-I mice (Fig. 4C). CD8/CD4 ratio was as high in AB*ab*-I mice as in C57BL6/N mice and 3-times higher than in AB*ab*-A2 mice (Fig. 4D).

Comparison of CD8⁺ and CD4⁺ T cell populations in lymphoid organs (on CD3⁺ cells) showed 12% of CD8⁺ T cells in AB*ab*-A2 with 62% of CD4⁺ T cells, and 47% of CD8⁺ T cells in AB*ab*-I with 39% of CD4⁺ T cells (Fig. 4E and Supplementary Fig. 4D). On average, AB*ab*-I mice had 3.5-times more absolute CD8⁺ T cell numbers in the major secondary lymphoid organs (Fig. 4F) with no significant difference in absolute CD4⁺ T cell numbers (Fig. 4G). As seen in peripheral blood, CD8/CD4 ratio was 2.5-times higher in AB*ab*-I mice compared to AB*ab*-A2 mice (Fig. 4H).

Unlike classical transgenic mice, AB*ab*-I mice were generated using the PB transposon technology. Though it produces single-copy integrations at a specific location, eventually, PB allows multiple-copy integrations on different chromosomes[19]. Due to this, T cell counts varied from mouse to mouse and decreased through F1–F5 generations, probably due to copy number loss. To understand and stabilize high T cell counts for efficient TCR discovery, we analyzed integration sites using targeted locus amplification (TLA) technology in F5 mice (Supplementary Fig. 5). Integration of the six-HLA genes cassette were detected on chromosomes 4, 7, 12, and 14. Next, we monitored T cell numbers over the generations F6-F10 and bred high CD8⁺ T cell counts mice with each other (Supplementary Figs. 6A, B and 4C). By this, we stabilized higher CD8⁺ T cell numbers in the periphery (Supplementary Fig. 6B) and created a functionally stable AB*ab*-I line.

### Broad CD8⁺ T cell repertoire in AB*ab*-I mice

The TCR repertoire was determined by a multiplex PCR. Therefore, genomic DNA of purified CD8⁺ T cells from pooled spleen and lymph nodes was isolated (Supplementary Fig. 4D). The genomic DNA equivalent of $3 \times 10^5$ (average in mice) CD8⁺ T cells per mouse was deep sequenced. Five mice per group were analyzed. Computational analysis was performed by ImmunoSEQ analyzer, and for the identification of specific V, D, and J genes, the International Immunogenetics (IMGT) nomenclature was used. In AB*ab*-I mice, TCRα repertoire deep sequencing quantified $11.8 \pm 1.4 \times 10^4$, and similarly, TCRß sequencing quantified $11.2 \pm 2.7 \times 10^4$ numbers of unique in-frame amino acid (aa) clonotypes in the CD8⁺ T cells (Fig. 5A and D). In AB*ab*-A2 mice, TCRα and TCRß sequencing quantified between $4.8 \pm 0.8$ and $5.2 \pm 0.8 \times 10^4$ numbers of unique aa clonotypes in the CD8⁺ T cells. Total valid reads for TCRα sequencing were between $0.34 \times 10^6$ and $0.91 \times 10^6$ for AB*ab*-A2 and AB*ab*-I, respectively. Valid reads for TCRß sequencing were between $0.29 \times 10^6$ and $0.45 \times 10^6$, respectively (Supplementary Table 1). AB*ab*-I mice had 7% of rare TCRα and 8.6% of rare TCRß clones, whereas AB*ab*-A2 had 5.2% of rare TCRα and 5.7% of rare TCRß clones (Fig. 5B and E). However, the frequency of small size clonotypes had increased in the TCRα (3.5%) and TCRß (4%) repertoire of AB*ab*-A2 compared to AB*ab*-I mice with 1.9% TCRα and 1.4% TCRß clones (Fig. 5B and E).

TCR sequencing encompasses only part of the complete repertoire of an individual. The estimation of the total TCR repertoire of CD8⁺ T cells in mice was determined by computational statistics. The iChao1 estimator calculated the total number of clonotypes per mouse (observed and undetected) using information from clones that occurred only once or twice[20]. iChao1 estimates true species richness using lower bound rarely occurring clones. AB*ab*-I mice had a diverse

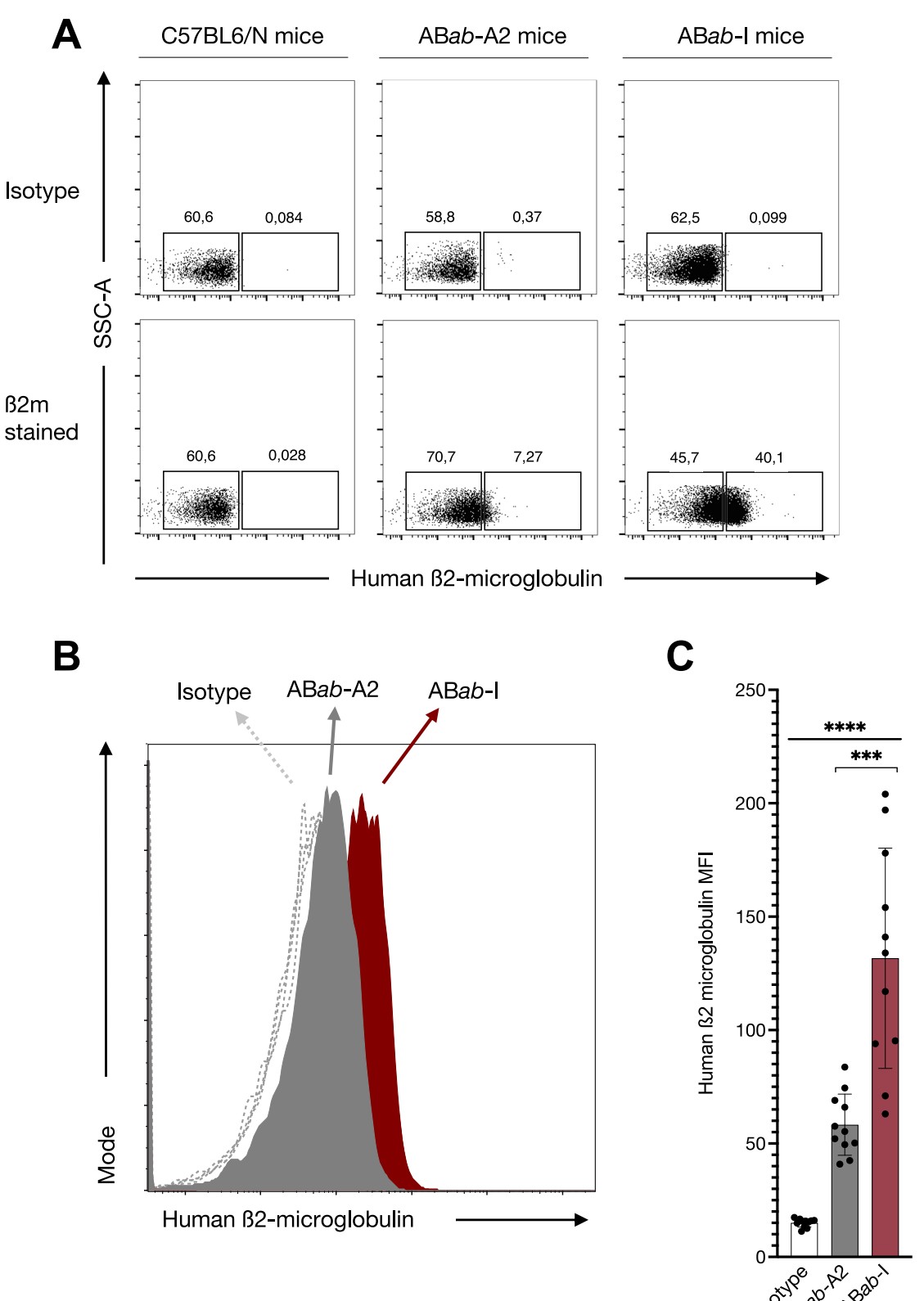

**Fig. 3 | Human HLA class I haplotype expression in AB*ab*-I mice. A** HLA staining of peripheral blood cells from C57BL6/N, AB*ab*-I, and AB*ab*-A2 mice with ß2m antibody, gated on lymphocytes. One representative flow cytometry plot out of multiple experiments is shown (*n* > 10). **B** Representative histogram overlay plot shows MFI comparison of ß2m stained HLA molecules among AB*ab*-A2 and AB*ab*-I mice. **C** ß2m MFI compared between AB*ab*-A2 (*n* = 11) and AB*ab*-I mice (*n* = 11). Data in (**C**) are presented as mean ± SD. Multiple group statistical comparison was performed using one-way ANOVA, followed by pairwise comparison between AB*ab*-A2 and AB*ab*-I groups using two-tailed unpaired Student's *t*-test. *P* values indicate: ****P* < 0.001; *****P* < 0.0001. Exact *P* values and source data are available in the Source Data file.

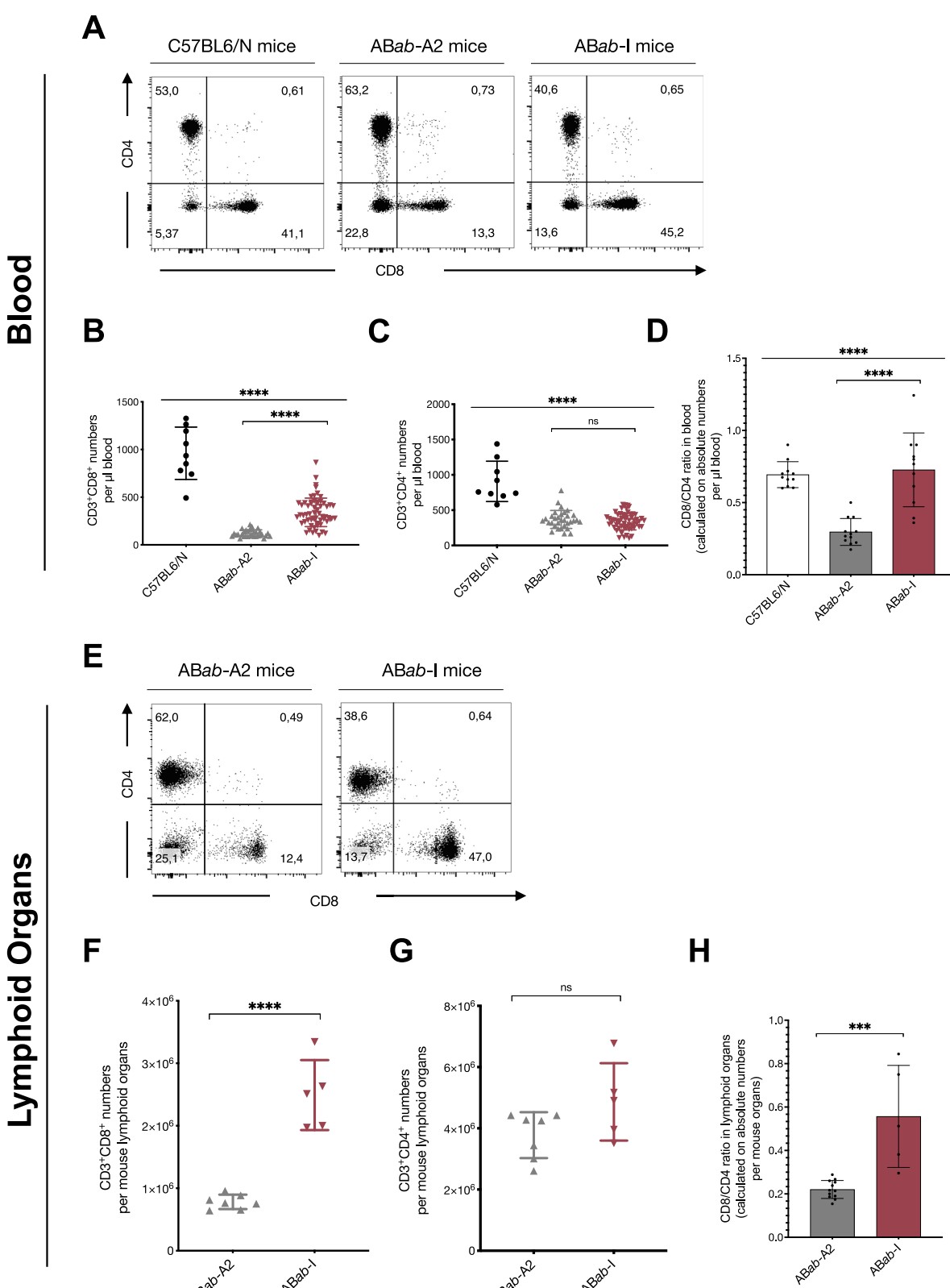

repertoire above AB*ab*-A2 mice with $2 \times 10^5$ TCRα and $4.4 \times 10^5$ TCRß clonotypes on average ($n = 5$). Using iChao1, we showed that AB*ab*-I mice used diverse V and J genes and produced close to $2.3 \times 10^5$ TCRα and $6.2 \times 10^5$ TCRß clonotypes compared with AB*ab*-A2. AB*ab*-A2 pool generated a maximum of $1.2 \times 10^5$ TCRα and $1.8 \times 10^5$ TCRß clonotypes ($\sim 2$–$3\times$ times lower), with an average of $0.89 \times 10^5$ TCRα and $0.95 \times 10^5$ TCRß clonotypes (Fig. 5C and F).

Vα (Fig. 6A) and Vß (Fig. 6B) gene usages were analyzed in both out-of-frame and in-frame TCR-α and -ß clonotypes. The former represents the preselection pool, which is an unbiased measure of the frequency by which a given V(D)J recombination event had occurred before selection. In both, AB*ab*-A2 and AB*ab*-I mice, most Vα and Vß segments were found to be rearranged, except for *TRAV1-1* in the case of α-chain and *TRBV5-1* and *TRBV6-1* in ß-chain, which were previously

**Fig. 4 | Phenotypic characterization of peripheral T cells in ABab-I mice.**
**A** Peripheral blood cells from young (8–12 weeks) indicated mouse strains were stained with antibodies specific for CD3, CD8, and CD4 chains and analyzed by flow cytometry (gated on CD3+ lymphocytes). One representative plot out of multiple experiments is shown (n > 30). **B** Absolute CD3+CD8+ T cell numbers/μl blood and (**C**) Absolute CD3+CD4+ T cell numbers/μl blood. Summarized data from C57BL6/N (n = 9), ABab-A2 (n = 32), and ABab-I (n = 60) mice. **D** CD8/CD4 ratio based on absolute numbers in peripheral blood from C57BL6/N (n = 11), ABab-A2 (n = 12), and ABab-I (n = 11) mice. **E** Spleen and lymph nodes from young (8–12 weeks) indicated mice were stained with antibodies specific for CD3, CD8, and CD4 chains and analyzed by flow cytometry (gated on CD3+ lymphocytes). One representative plot

out of multiple experiments is shown (n > 10). **F** Absolute CD3+CD8+ T cell numbers/mouse and (**G**) Absolute CD3+CD4+ T cell numbers/mouse. Summarized data from ABab-A2 (n = 7) and ABab-I (n = 5) mice. **H** CD8/CD4 ratio based on absolute numbers in lymphoid organs from ABab-A2 (n = 12) and ABab-I (n = 5) mice. Each data point in scatter plots represents an individual mouse from indicated strains. Data in (**B**–**D**), (**F**–**H**) are presented as mean ± SD. Multiple group statistical comparisons in (**B**–**D**) were assessed by one-way ANOVA, followed by pairwise comparisons between ABab-A2 and ABab-I groups using two-tailed unpaired Student's t-test. P values indicate: ***P < 0.001 and ****P < 0.0001; ns not significant. Exact P values and source data are available in the Source Data file.

reported for non-existence in the primary ABab strain or lack of expression (Fig. 6A, B)[7]. ABab-A2 and ABab-I mice showed no significant difference among any preferred usage of $V_\alpha/V_\beta$ in the pre-selection pool. As reported previously[15], V gene usage was far from random. In the pre-selection pool, recombination frequency of a number of $V_\alpha$ and $V_\beta$ genes were either over or underrepresented in both, ABab-A2 and ABab-I mice.

The post-selection V gene usage mirrored to a large extent that of the pre-selection pool. However, frequency of V gene usage was also HLA-dependent. Preferential overrepresentation of *TRAV12-02*, *TRAV17*, *TRAV22* or *TRBV12-03/12-04* in ABab-A2 was observed compared to ABab-I mice (Fig. 6A, B). Conversely, *TRBV04-01* was found overrepresented in ABab-I compared to ABab-A2 mice (Fig. 6B).

$J_\alpha$ and $J_\beta$ gene usages were also non-random and similar among the two transgenic mice (Supplementary Figs. 7 and 8). We detected almost all V-J gene combinations in both the α chain and the β chain of the in-frame and the out-of-frame repertoire (Supplementary Fig. 7). We noticed distinct differences in the V-J pairing in TCR selection between ABab-I and ABab-A2 mice. However, the post-selection pool for both α and β chain repertoire was similar to the pre-selection pool, except for some variations near the 5' end of TRAV-AJ and TRBV-BJ pairing (Supplementary Fig. 8).

### T cell receptors with longer CDR3 region in ABab-I mice
ABab-I mice had significantly longer CDR3α (42 <bp <51) and CDR3β (45 <bp <51) chain lengths compared to ABab-A2 mice, which expressed TCRs with shorter CDR3α (33 <bp <39) and CDR3β (33 <bp <42) lengths (Fig. 7). In ABab-A2 mice, 6% and 5.5% of clonotypes expressed TCRs with 33 bp, 13% and 11.5% with 36 bp, and 20% and 21% with 39 bp CDR3 α and β lengths, respectively. In ABab-I mice, lengths corresponded to 5% and 4.9% with 33 bp, 12% to 10% with 36 bp, and 19.6% and 19.1% with 39 bp CDR3 α and β lengths, respectively (Fig. 7A and C). In ABab-I mice, on average, more clones expressed TCRs with longer CDR3β regions compared to ABab-A2. In CDR3β, 23% versus 21% with 45 bp, 10% versus 8.3% with 48 bp, and 3.7% versus 3.4% with 51 bp (Fig. 7C). ABab-I mice had TCRs with 42 ± 0.1 bp CDR3β regions compared with 41.5 ± 0.06 bp in ABab-A2 mice (Fig. 7D). Average CDR3α regions were similar between ABab-A2 (41.6 ± 0.06 bp) versus ABab-I mice (41.9 ± 0.04 bp), respectively (Fig. 7B).

### High numbers of unique clones in ABab-I mice
Shared clones were compared between the two mouse strains ABab-I and ABab-A2 using Jaccard similarity index estimation. Jaccard index (J) evaluates the shared immune repertoire between samples. J index ranges from 0 to 1, and the score is calculated based on the formula: the total number of shared clones divided by the total number of unique clones across two samples, $J(A, B) = (A \cap B)/(A \cup B)$.

Higher shared clonality within strains (ABab-I with ABab-I or ABab-A2 with ABab-A2) were detected than across strains (ABab-I with ABab-A2 or vice versa). ABab-I mice shared ~ 17% of TCRα clones among each other, Jaccard index: 0.17 ± 0.005, and ABab-A2 mice shared ~ 13% of clones within their group, Jaccard index: 0.13 ± 0.002 (Fig. 8A and Supplementary Fig. 9A). Number of shared TCRα

clonotypes within each group was higher than the number of shared TCRβ clones due to single versus two somatic recombination events. ~ 6.4% of TCRβ clones were shared among ABab-I mice, Jaccard index: 0.0636 ± 0.003, and ~ 5% of clones within the ABab-A2 group, Jaccard index: 0.0498 ± 0.004 (Fig. 8C and Supplementary Fig. 9B).

ABab-I mice generated more unique clones compared to ABab-A2 mice. Pooled analysis from 5 mice per strain determined the total number of shared and unique clones. ABab-A2 and ABab-I mice shared <100,000 TCRα and TCRβ clones (0.8 and $0.5 \times 10^5$, respectively) between each other. ABab-I mice produced almost 3-times more unique and rare clones (ABab-I: $3.0 \times 10^5$ α clonotypes and $4.2 \times 10^5$ β clonotypes) than ABab-A2 mice ($1.0 \times 10^5$ TCRα and $2 \times 10^5$ TCRβ clones) (Fig. 8B and D).

### All six different HLA-I molecules present peptides ex vivo
To determine epitope presentation by the six novel HLA-I alleles in ABab-I mice, peripheral blood or lymphoid organ cells were pulsed with peptides, which were known to be presented by the respective HLA-I molecule, and co-cultured with T cells transduced with TCRs which were specific for these epitopes. Previously described (A3-, A11-, B7-, B15-, and C7-restricted), or in-house generated (C4-restricted) TCRs against a panel of antigens[21–25] were used in the co-culture assays (Fig. 9A and Supplementary Fig. 10).

Co-culture supernatants were analyzed for murine IFN-γ to assess effector T cells for recognition of the respective peptide-HLA molecules. No IFN-γ production was detected when three (A11-, C7-, and A2-) TCRs were tested in the absence of their cognate peptides, confirming the specificity of TCR recognition (Supplementary Fig. 10). No response was observed for A3-, B7-, B15-, and C4-TCRs in the absence of peptides[21,23,24]. Upon recognition of peptide-bound HLAs, all six TCR-transduced T cells produced IFN-γ in a cognate HLA-peptide-TCR manner, proving functional activity (Fig. 9B).

### ABab-I mice mount effective immune responses against tumor antigens in vivo
To analyze antigen-specific CD8+ T cell responses against tumor antigens in vivo, ABab-I mice were immunized with peptide antigens. All six antigens were in silico predicted to be strong-binders to their cognate HLA alleles (Fig. 9A).

Intracellular staining measured 1.8% of CD8+ T cells producing IFN-γ against the mutated calreticulin (mCALR) epitope, KMRMRRMRR. 6.9% of CD8+ T cells released IFN-γ against the mutated KRAS G12V epitope, VVGAVGVGK. 1.4% against *B*07:02* bound CMV epitope, TPRVTGGGAM, 0.2% against *B*15:01* bound HPV epitope, SAFRCFIVY, 0.3% against *C*04:01* bound CMV epitope, QYDPVAALF, and 0.23% against *C*07:02* bound MAGE-A12 epitope, VRIGHLYIL (Fig. 10 and Supplementary Fig. 4E). CD8+ T cell responses towards a panel of epitopes in the ABab-I mouse model depicted the in vivo functionality of the human HLA haplotype.

## Discussion
The ABab-I transgenic mouse model developed in this study addresses a key challenge in advancing adoptive T cell therapy by

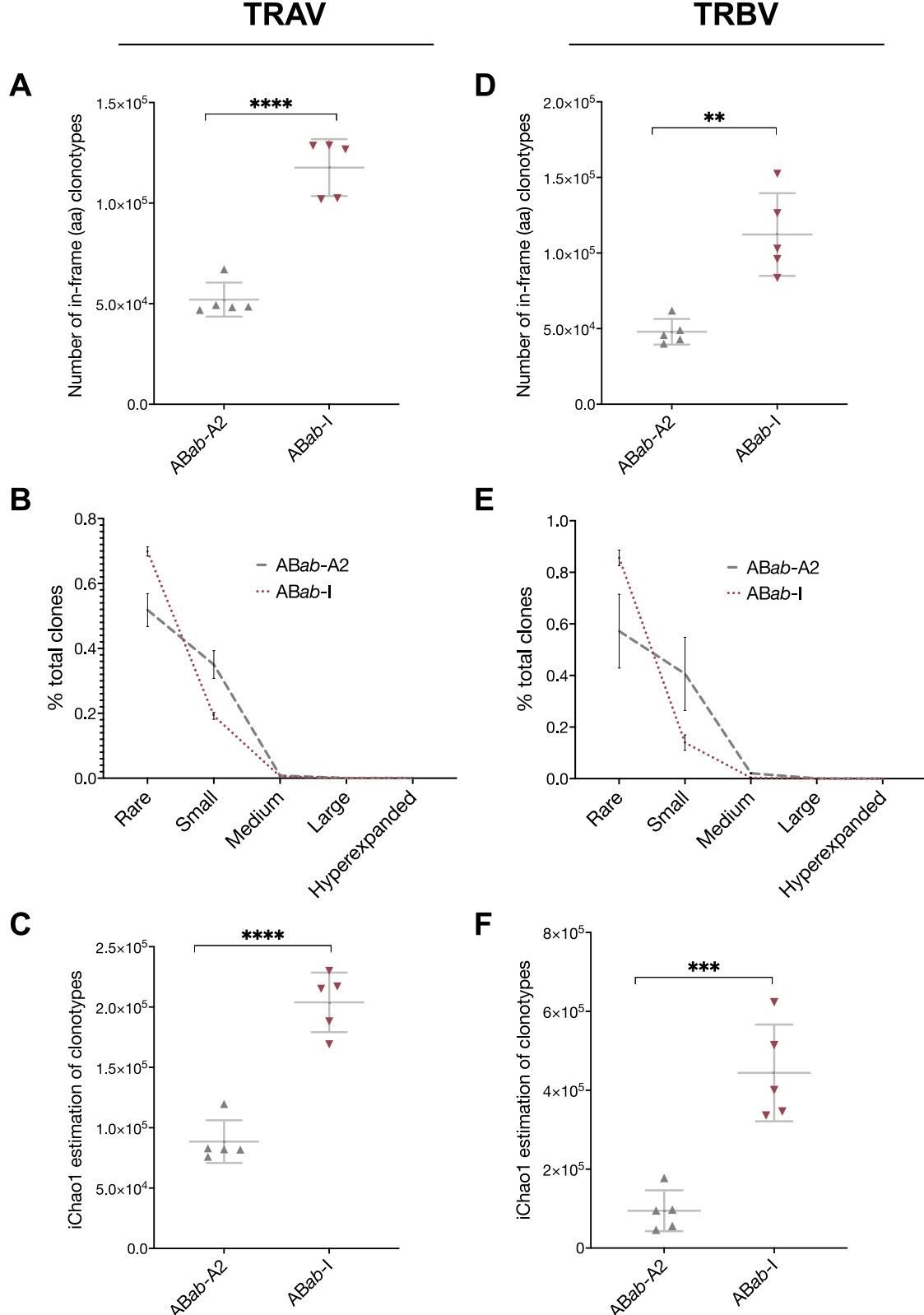

**Fig. 5 | TCRαβ repertoire in AB*ab*-I mice. A** and **D** Absolute numbers of unique TCRα (TRAV) and TCRß (TRBV) amino acid (aa) clonotypes within $3 \times 10^5$ CD8+ T cells (average in mouse strains). **B** and **E** Clonal distribution of TCRα and TCRß aa clonotypes of different sizes: rare, $0 < x \le 0.001\%$; small, $0.001\% < x \le 0.01\%$; medium $0.01\% < x \le 0.1\%$; large, $0.1\% < x < 1\%$; hyperexpanded, $1\% < x < 10\%$. **C** and **F** TCRα and TCRß diversity was calculated using the iChao1 estimator (lower bound richness of total numbers of unique templates within an individual repertoire). Data are from AB*ab*-A2 ($n = 5$) and AB*ab*-I ($n = 5$) mice. Data in (**A**), (**C**), (**D**), and (**F**) are presented as mean ± SD. Pairwise statistical comparisons between the AB*ab*-A2 and AB*ab*-I mouse groups were performed using two-tailed unpaired Student's *t*-test: **$P < 0.01$, ***$P < 0.001$, and ****$P < 0.0001$. Exact *P* values and source data are available in the Source Data file.

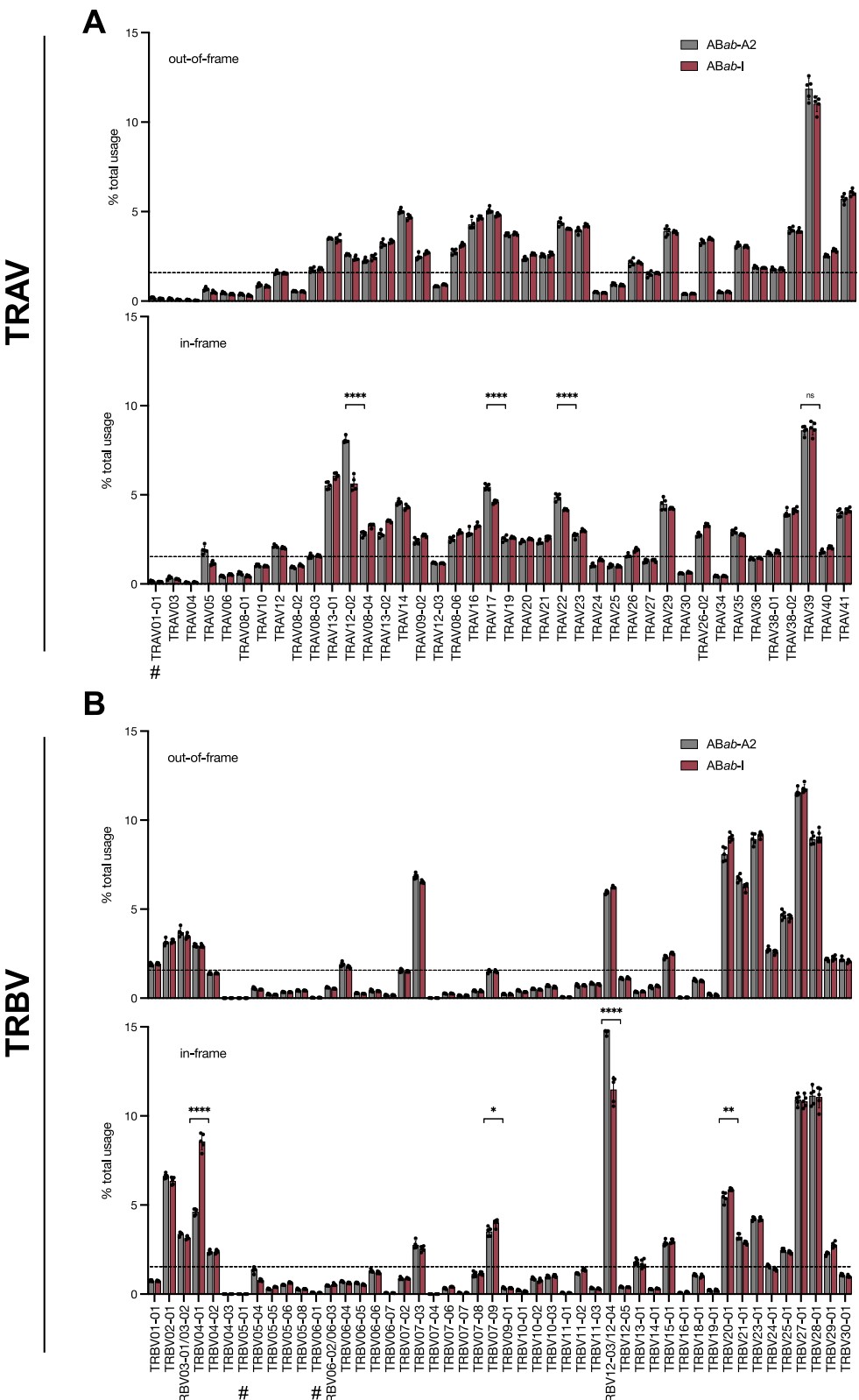

**Fig. 6 | Vα and Vβ gene usage frequencies of in-frame and out-of-frame TCRαβ clonotypes.** Frequencies of Vα (**A**) and Vβ (**B**) gene usages of unique TCRαβ clonotypes in CD8⁺ T cells of AB*ab*-A2 and AB*ab*-I mice. The dotted line represents the frequency of random Vαβ gene usage (TRAV, 2.3%; TRBV, 2.1%). Arrangement of Vα and Vβ gene segments mentioned on the *x*-axis according to their position on the human chromosome from 5′ to 3′. 1st top lane: out-of-frame and 2nd bottom lane: in-frame frequencies. #Expression of Vαβ genes is missing in both AB*ab*-A2 and AB*ab*-I mice. Data are from AB*ab*-A2 (*n* = 5) and AB*ab*-I (*n* = 5) mice. Data in (**A**) and (**B**) are presented as mean ± SD. Pairwise statistical comparisons between the AB*ab*-A2 and AB*ab*-I mouse groups were performed using two-tailed unpaired Student's *t*-test: \**P* < 0.05, \*\**P* < 0.01, and \*\*\*\**P* < 0.0001; ns not significant. Exact *P* values and source data are available in the Source Data file.

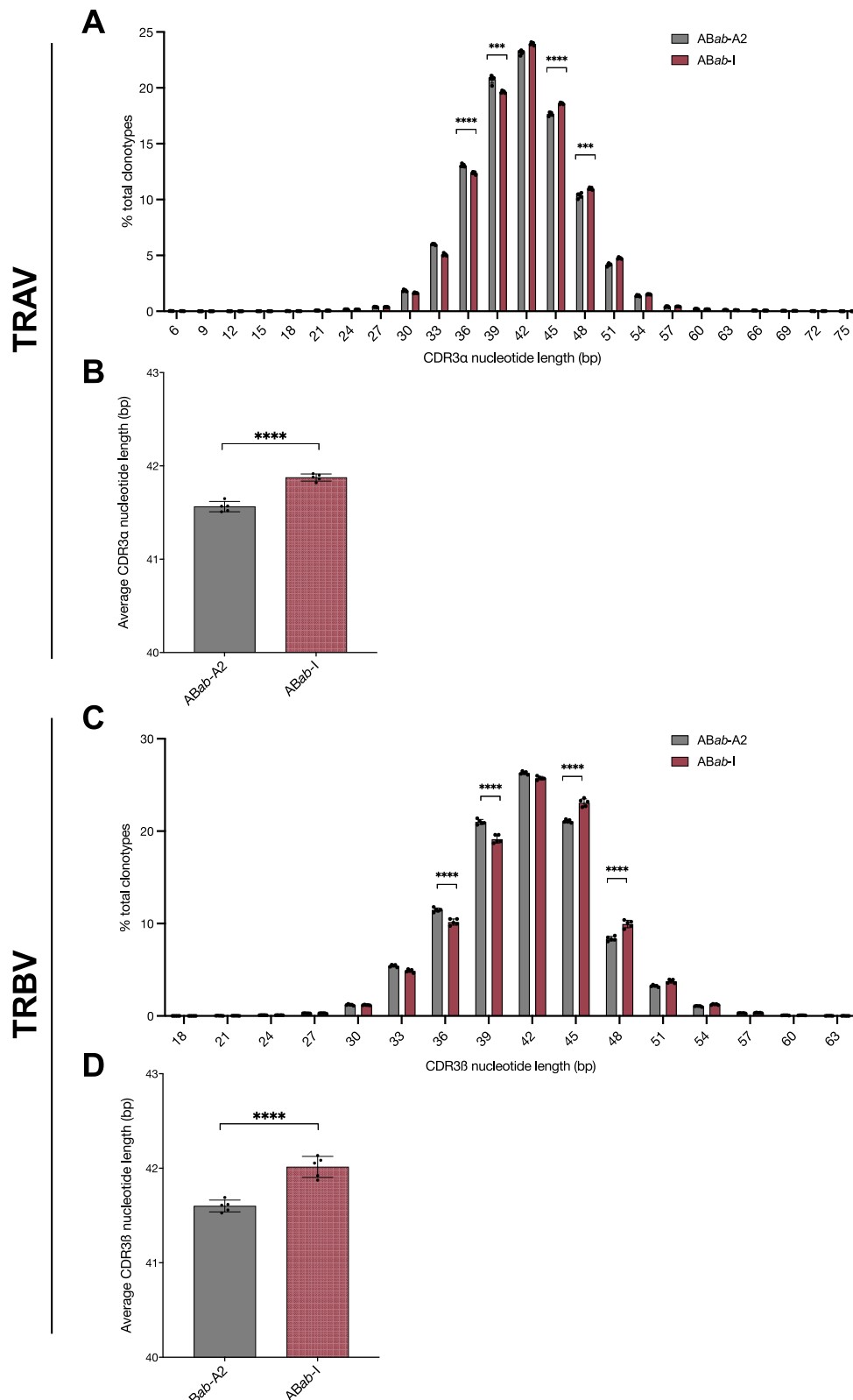

**Fig. 7 | CDR3 region analysis in AB*ab*-A2 and AB*ab*-I mice.** CDR3α (**A**) and CDR3ß (**C**) length distributions. In-frame frequencies of different CDR3α (**A**) and ß (**C**) lengths for AB*ab*-A2 and AB*ab*-I mice is shown in the distribution graphs. Average CDR3α (**B**) and ß (**D**) lengths were compared between AB*ab*-A2 and AB*ab*-I mice. Data are from AB*ab*-A2 (*n* = 5) and AB*ab*-I (*n* = 5) mice. Data in panels are presented as mean ± SD. Pairwise statistical comparisons between the AB*ab*-A2 and AB*ab*-I mouse groups were performed using two-tailed unpaired Student's *t*-test: ***$P$ < 0.001 and ****$P$ < 0.0001. Exact $P$ values and source data are available in the Source Data file.

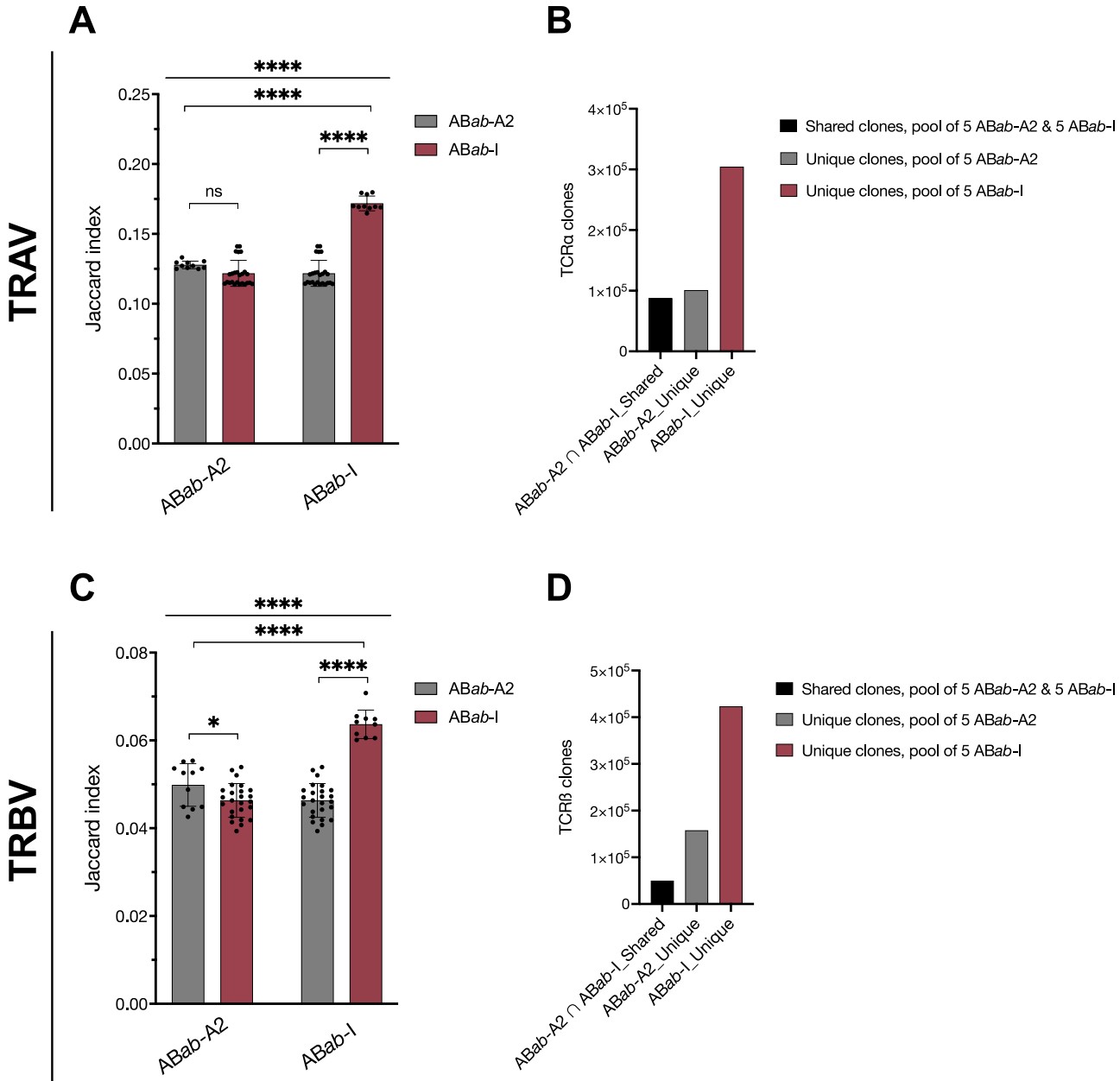

**Fig. 8 | Shared and unique clones among AB*ab*-A2 and AB*ab*-I mice. A** and **C** Jaccard index represents a score based on the number of shared TCRα and ß clones within and between groups of AB*ab*-A2 and AB*ab*-I mice. **B** and **D** Total numbers of unique TCRα and TCRß clones in AB*ab*-A2 and AB*ab*-I from pooled data. Five mice per strain were pooled for analysis. Data are from AB*ab*-A2 (*n* = 5) and AB*ab*-I (*n* = 5) mice. In (**A**) and (**C**), each data point represents a pairwise comparison between individual mice within or across groups, resulting in multiple data points per comparison category (i.e., per bar). Data in (**A**) and (**C**) are presented as mean ± SD. Multiple group statistical comparisons in (**A**) and (**C**) were performed by one-way ANOVA, followed by pairwise comparisons using two-tailed unpaired Student's *t*-test. *P* values indicate: **P* < 0.05 and *****P* < 0.0001; ns not significant. Exact *P* values and source data are available in the Source Data file.

overcoming the limitations of human-derived TCRs, which often exhibit low avidity, hindering clinical translation. In contrast, transgenic models like the novel AB*ab*-I mice provide a naïve, non-tolerant TCR repertoire that, upon immunization, can generate high-affinity TCRs suitable for therapeutic use[7,11,12]. The hyperactive PiggyBac transposase introduced six human HLA alleles in previously existing humanized TCR mice[7], resembling a complete human HLA class I haplotype-TCR recognition system. The PiggyBac transposon technology proved to be efficient but bears the risk of losing copy numbers of multiple chromosomal integration sites, likely due to the gradual loss of integrations over successive breeding generations rather than transposon excision. Therefore, the breeding strategy was based on selecting mice with high peripheral CD8+ T cell counts as sign of efficient positive selection and in turn efficient expression of the multiple HLA class I genes in both, the thymus and peripheral lymphoid organs. Since generation F8, the AB*ab*-I line was stable and yielded uniformly high CD8+ cell numbers in the periphery. Afterwards, we have propagated the line for more than 10 generations with continuously high and stable peripheral CD8+ cell numbers. Of note, the TCR repertoire analysis was performed with "interim" mice (CD8+ T cell numbers shown in Fig. 4), and not the final AB*ab*-I mice (shown in Supplementary Fig. 6). Thus, the diversity of the TCR repertoire presented here is likely an underestimate. The six HLA class I genes HLA-*A*03:01*, *-A*11:01*, *-B*07:02*, *-B*15:01*, *-C*04:01*, and *-C*07:02* were chosen due to their high frequency among the world population.

## A

| Model TCRs restricted towards epitopes presented by HLA alleles in AB*ab*-I mice | | | | |
|---|---|---|---|---|
| HLA restriction | TCR chains | V(D)J recombined regions | CDR3 regions | Epitope sequences (IC$_{50}$) |
| A*03:01 | TCR Vα chain<br>TCR Vβ chain | TRAV9-2-TRAJ30-1<br>TRBV13-1-TRBD2-1-TRBJ2-3 | CALSDRERDDKIIF<br>CASSHEGLAGEFF | KMRMRRMRR (29.18 nM)<br><br>*mCALRp7 neoantigen* |
| A*11:01 | TCR Vα chain<br>TCR Vβ chain | TRAV3-3-TRAJ17-1<br>TRBV4-1-TRBD2-1-TRBJ2-1 | CAVSGGTNSAGNKLTF<br>CASSRDWGPAEQFF | VVGAVGVGK (65.47 nM)<br><br>*KRAS G12V neoantigen* |
| B*07:02 | TCR Vα chain<br>TCR Vβ chain | TRAV17-1-TRAJ12-1<br>TRBV7-9-TRBD1-1-TRBJ2-1 | CATVIRMDSSYKLIF<br>CASSLIGVSSYNEQFF | TPRVTGGGAM (3.86 nM)<br><br>*CMV pp65 viral antigen* |
| B*15:01 | TCR Vα chain<br>TCR Vβ chain | TRAV17-1-TRAJ47-2<br>TRBV6-5-TRBD1-1-TRBJ2-5 | CAESEYGNKLVF<br>CASSYRQQETQYF | SAFRCFIVY (118.20 nM)<br><br>*HPV16 E5 viral antigen* |
| C*04:01 | TCR Vα chain<br>TCR Vβ chain | TRAV26-2-TRAJ49-1<br>TRBV15-2-TRBD2-2-TRBJ2-1 | CILRDNTGNQFYF<br>CATSRDGSSYNEQFF | QYDPVAALF (499.34 nM)<br><br>*CMV pp65 viral antigen* |
| C*07:02 | TCR Vα chain<br>TCR Vβ chain | TRAV13-1-TRAJ34-1<br>TRBV25-1-TRBD2-1-TRBJ2-7 | CAASGATDKLIF<br>CASSGGHEQYF | VRIGHLYIL (647.00 nM)<br><br>*MAGE-A12 shared antigen* |

## B

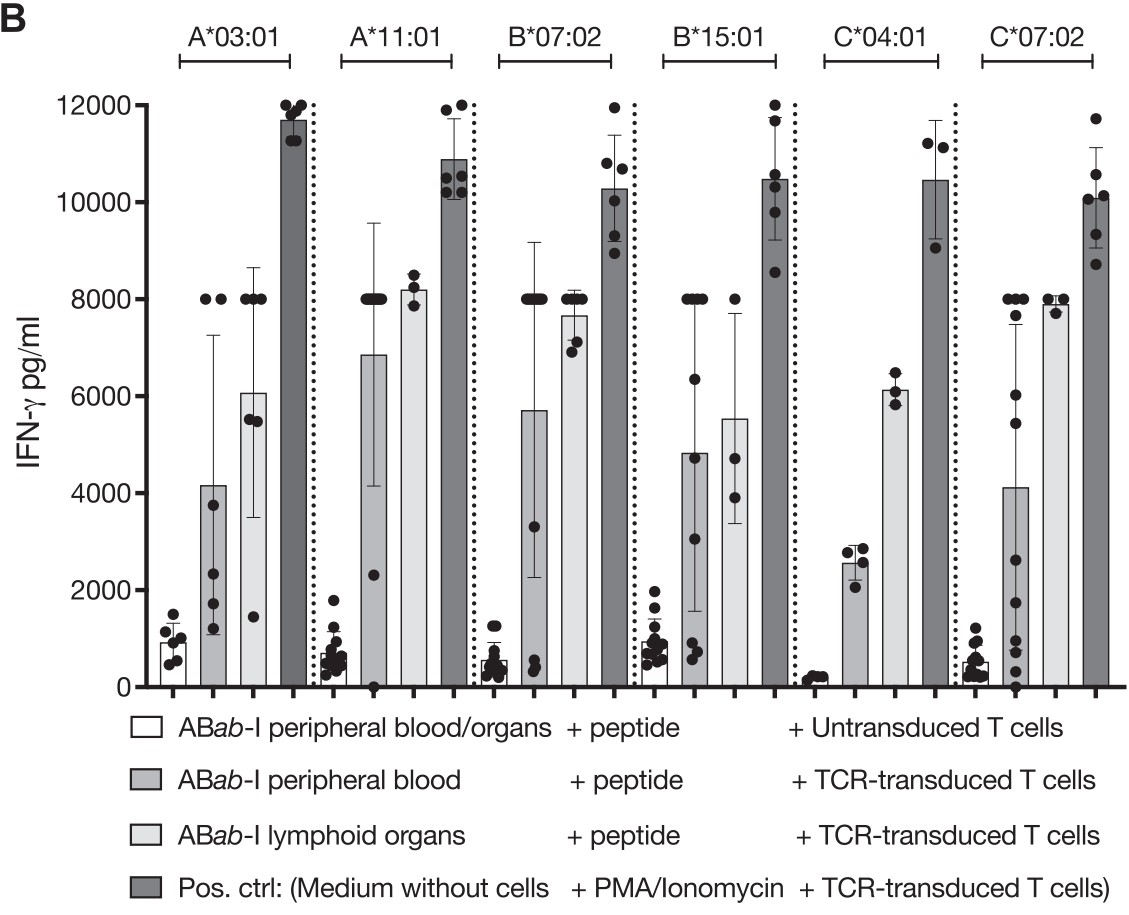

**Fig. 9 | Functional activity of HLA molecules in AB*ab*-I mice. A** List of TCRs restricted towards HLA molecules in AB*ab*-I mice with known peptide binders were used in presentation assays. **B** Ex vivo-isolated peripheral blood cells or cells from lymphoid organs from AB*ab*-I mice, co-cultured with TCR-transduced or untransduced T cells. Cytokine levels were measured after 16 h of co-culture by mouse IFN-γ ELISA assay. Consolidated data from multiple TCR co-culture experiments are shown, grouped by HLA restriction and ex vivo tissue source as follows: A*03:01, *n* = 6 (peripheral blood), *n* = 6 (lymphoid organs); A*11:01, *n* = 12 and 3; B*07:02, *n* = 12 and 6; B*15:01, *n* = 10 and 3; C*04:01, *n* = 4 and 3; C*07:02, *n* = 12 and 3, respectively. Data in (**B**) are presented as mean ± SD. Source data are available in the Source Data file.

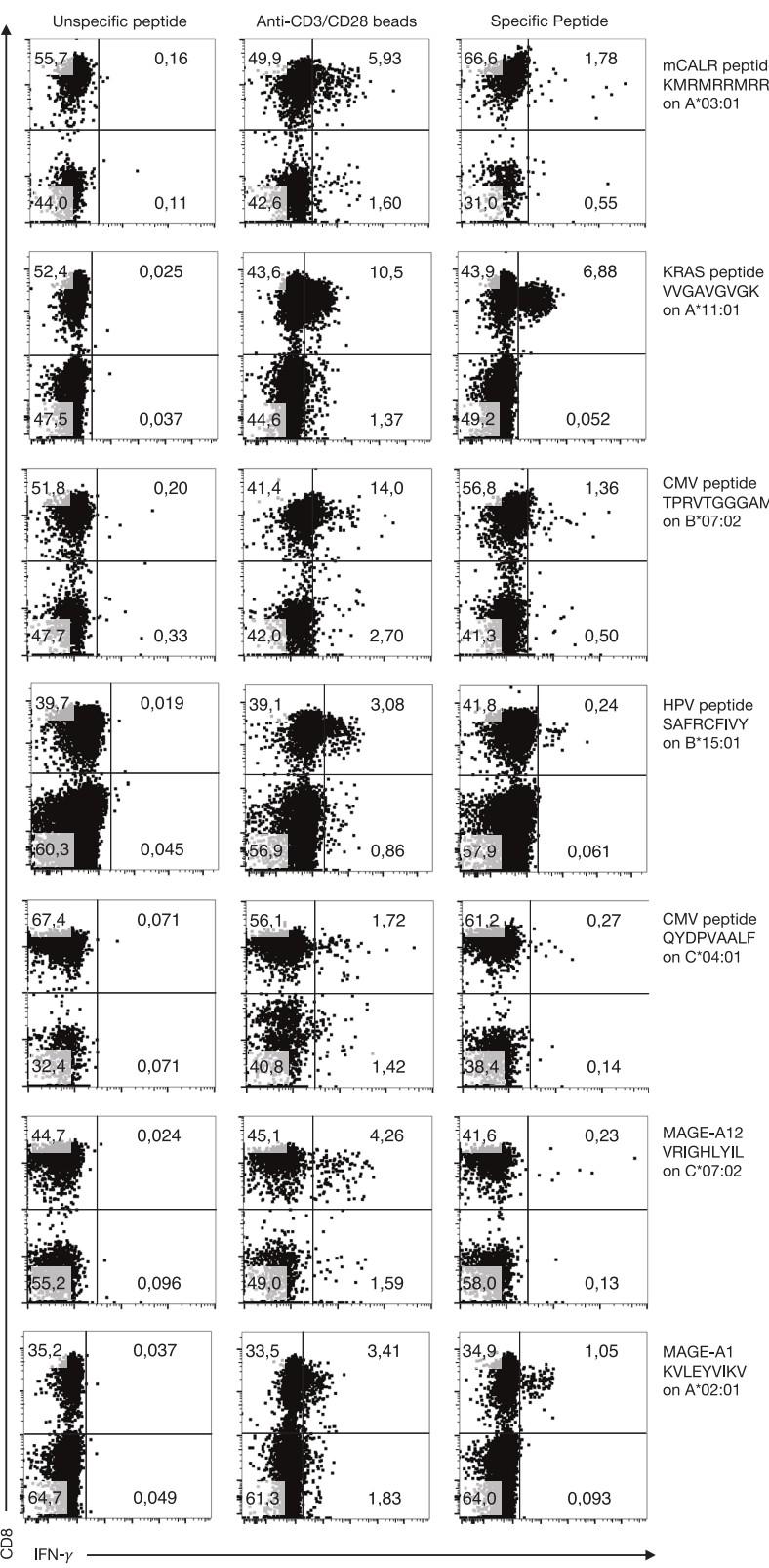

**Fig. 10 | CD8⁺ T cell responses against human tumor antigens in AB*ab*-I mice.**
Specific CD8⁺ T cell responses in AB*ab*-I mice were observed after immunization
with a panel of antigens. AB*ab*-I mice were immunized (minimum twice) with an
interval of at least 4 weeks between peptide injections. Seven days post-injection,
peripheral blood cells were stimulated in vitro with the same specific peptides used
for immunization (right), or unspecific NY-ESO peptide, APRGPHGGAASGL (left), or
CD3/CD28 beads (middle, pos. control). Cells were analyzed for CD3, CD8, and
intracellular IFN-γ expression by flow cytometry. Dot plots depict CD8⁺ IFN-γ⁺
T cells (gated on CD3⁺ lymphocytes). One representative plot out of multiple
responders is given (out of 6–7 mice/antigen, 6 mice responded against KRAS, 3
mice against HPV and MAGE-A1, 2 mice responded against CMV (1 each to
TPRVTGGGAM and QYDPVAALF peptides), 1 mouse responded against mCALR and
MAGE-A12 peptides).

Expression of multiple class I HLAs improved positive selection in the thymus to generate large numbers of CD8[+] T cells in the periphery. AB*ab*-I mice had normal homeostatic behavior in the peripheral CD8[+] T cell compartment, which rescued from lymphopenic condition that existed in AB*ab*-A2 mice, which express a single HLA-I gene[7,26,27]. In AB*ab*-I mice, increased CD8[+] T cell numbers broadened the TCR repertoire with a higher percentage of unique clones for an unbiased TCR discovery against human tumor antigens. The multiple class I alleles in AB*ab*-I mice will allow to analyze the relative immunogenicity of different epitopes of a given antigen presented by the different HLA class I molecules. As the focus has been on HLA-A2 presented peptides for a long time, AB*ab*-I mice may allow to discover novel epitopes as well as respective TCRs, especially for human TAA, which are foreign to the mice. In contrast to mice, which are often negative for human TAA, humans are typically tolerant and bear only low-affinity TCRs. TCRs specific for TAA bear the risk of exhibiting off-target reactivity and toxicity, because these TCRs were not negatively selected against human antigens. Due to the protein sequence discrepancy between human and mice, there may be potential cross-reactivity to human self-antigens. Therefore, we previously employed a series of experiments with the aim to minimize the risk of off-target toxicity prior to clinical translation. These experiments included alanine scan to determine the TCR contact residues of the peptide, searching the human proteome for peptides with the same TCR contact motif and testing these peptides for recognition by the TCR under investigation. Furthermore, we tested the TCRs for recognition of a library of self-peptides and, for testing HLA alloreactivity, a library of human LCL cell lines covering a large number of MHC-I and MHC-II alleles[11,12].

Following immunization with different peptide antigens known to be presented by any of the six HLA-I alleles, AB*ab*-I mice generated efficient CD8[+] T cell responses, proving that all six novel alleles were functional. We introduced the six HLA class I genes into AB*ab*-A2 mice, so they in fact contained seven HLA class I genes, in order to avoid excessive breeding to delete mouse TCR and mouse MHC class I expression.

Apart from AB*ab*-I mice being a useful tool for epitope and TCR discovery, they are also interesting with regard to T cell development. It has been hypothesized that the TCR and the MHC genes coevolved for an intrinsic binding, rendering positive selection more efficient. Elegant work supported this hypothesis, but other work argued against[28,29]. Indeed, it is difficult to see how dozens of $V_\alpha$ and $V_\beta$ genes could have evolved for a high intrinsic affinity towards hundreds to thousands of HLA alleles. Because of the extreme HLA class I gene polymorphism one can assume that the intrinsic affinity for some $V_\alpha$ or $V_\beta$ gene segments is higher for some HLA-I alleles but lower for others[28]. This hypothesis was supported by Chen and colleagues[15]. They sequenced the human CD4[+] TCR repertoire in mice selected on either a single mouse or a single human HLA class II molecule and found that the species-compatible human HLA class II did select a broader CD4[+] TCR repertoire compared to the species-incompatible mouse MHC molecule *I-A^b*. This could best be explained by an increased inherent affinity of the human TCRs for the human HLA class II or, in other words, coevolution of human TCR and HLA gene loci after divergence from mice[15]. In analogy, one might assume that some V gene segments have a higher inherent affinity to HLA-A2, whereas others have a lower intrinsic affinity to HLA-A2, but in turn a higher inherent affinity for any of the other six HLA class I alleles in AB*ab*-I mice. This then would mean, that certain TCRs are less favorably selected on HLA-A2 in AB*ab*-A2 mice but are more efficiently selected by the other HLA-I alleles in AB*ab*-I mice.

Chen *et al.* also observed that TCRs selected on mouse MHC-II had, on average, shorter CDR3 sequences compared to human TCRs that were selected on human HLA-II molecules[15]. The hypothesis is that different inherent affinities of $V_\beta$ gene segments for MHC were adjusted by CDR3 lengths, i.e., a particular $V_\beta$ gene segments with a relatively low inherent affinity for HLA were better selected if they carried shorter CDR3, assuming that longer CDR3 decrease the inherent affinity. We extend these findings because a TCR repertoire selected on multiple class I alleles contained, on average, slightly larger CDR3β regions lengths compared to the TCR repertoire selected only on a single HLA class I allele, in our case HLA-A2. The best explanation is that some $V_\beta$ segments have an increased inherent affinity for any of the six novel MHC class I alleles in AB*ab*-I mice, which allows for the selection of TCRs with a slightly larger CDR3 region. This not only explained the highly increased diversity in AB*ab*-I compared to AB*ab*-A2 mice but could also result in TCRs with higher specificity. This assumption is based on studies in mice that were deficient in terminal deoxynucleotidyl transferase (*TdT*) expression, which resulted in TCRs with short CDR3s. These TCRs tended to be more cross-reactive[30]. Even though the TCR repertoire in AB*ab*-A2 mice is relatively diverse, we speculate that the TCR repertoire in AB*ab*-A2 mice is not optimal because the CDR3 regions are relatively short[15]. In contrast, thymocytes in AB*ab*-I mice can choose one out of six additional class I alleles for being selected. Therefore, we assume that different HLA alleles in the thymus compete for selecting a given T cell clone. If this is correct, T cell clones that are not being able to be selected by HLA-A2 could be selected by any of the other six HLA class I alleles for which they have an increased inherent affinity and then also allowing longer CDR3s being accepted by not impeding positive selection.

It should be noted, however, that the hypothesis that the T cells in AB*ab*-I mice find its HLA fit with optimal intrinsic affinity at a price of reduced selection on HLA-A2 remains speculative so far. This hypothesis is difficult to prove because it is also not known whether different HLA class I alleles can select similar diverse TCR repertoire. Therefore, we address this limitation by currently generating and characterizing mice with single HLA alleles of those that are contained in AB*ab*-I mice. Using these mice, we can prove or disprove whether certain human $V_\beta$ gene segments are preferentially selected or preferentially ignored by individual HLA-I alleles[15,31]. The question whether more HLA alleles have a positive or negative impact on TCR diversity is controversially discussed. Some data suggest that more HLA alleles reduce the TCR repertoire because of increased negative thymic selection, while other data suggest that an intermediate number of HLA alleles was associated with the largest T cell diversity[32–35]. The former was shown for CD4[+] T cells in mice[33], the latter for CD8[+] T cells in bank voles (*Myodes=Cletronomys glareolus*)[32]. In humans, there is little variation from the number of three HLA-I alleles per haploid genome, resulting in usually six different HLA-I genes because of evolutionary selection for heterozygocity. Our data demonstrate that the number of HLA-I alleles in AB*ab*-I mice, resembling the human situation, select by far more total CD8[+] T cells and a substantially broader TCR repertoire compared to AB*ab*-A2 mice with just one HLA-I allele. We hypothesize that evolution selected for six HLA-I alleles in humans to balance between optimal presentation of peptides from pathogens and selection of an optimally diverse TCR repertoire.

In conclusion, AB*ab*-I mice are a useful tool for tumor antigen epitope and TCR discovery. They also reveal that the post-selection TCR repertoire largely reflects the pre-selection TCR repertoire, that different HLA-I alleles compete for positive selection and that different inherent affinities of V segments for individual HLA-I alleles are adjusted by the CDR3 length for optimal positive selection.

## Methods

### Study design

The study aimed to develop transgenic mice with six human HLA alleles for TCR and tumor antigen epitope discovery and to study the effect of multiple HLAs on selecting a highly diverse human TCR repertoire in mice. To accomplish this, we first conducted in silico screens to identify top six class I HLAs based on their frequency in the Caucasian population and their potential high binding affinity of

recurrent cancer neoantigens. Next, we cloned the selected six HLA alleles in a single targeting vector similar to a natural human HLA class I haplotype situation to study the effect of multiple HLAs on selection of human TCRs in mice. All HLA alleles were chimeric human-mouse fusion proteins with α1 and α2 heavy chain cDNA regions from the respective human HLA genes, while α3, transmembrane, and cytoplasmic domains were from the murine *H-2 D^b* gene. PiggyBac ITR-flanked targeting vector carrying six HLA genes with each regulated by the H-2 promoter were co-injected with the hyperactive PiggyBac transposase mRNA into AB*ab*-A2 mouse oocytes. Pronuclei injection (PNI) yielded multiple AB*ab*-I founder lines. We performed genetic characterization, immune phenotyping, in vivo and ex vivo HLA expression analyses, immunization experiments with diverse antigens, and TCR repertoire deep sequencing to test the development and function of the CD8⁺ T cell compartment. We compared AB*ab*-I and AB*ab*-A2 mice to wild-type C57BL/6 N mice. In TCR deep sequencing, the iChao1 estimator calculated true species richness using lower bound rarely occurring clones. Throughout the study, we used 8- to 20-week-old mice for characterization experiments (at least 5 mice/group, 6–7 mice/group for immunizations). All mice were housed and maintained under specific pathogen-free (SPF) conditions, and experimental and control mice were placed in separate cages. We were granted permission to conduct all mouse experiments in this study by the Landesamt für Arbeitsschutz, Gesundheitsschutz und Technische Sicherheit, Berlin (Animal License Numbers: X9005/22, X9010/21, G-0111/17-16082, G-0111/17-16082-2, TVV-322/10), in accordance with the standard guidelines approved by the committee. At the end of animal experiments, the mice were euthanized by cervical dislocation.

### Transgenic mice generation
The six HLA alleles were designed as chimeric human-mouse fusion genes. The α1 and α2 domains were human cDNAs encoding the respective HLA alleles, and α3 to cytoplasmic domains were murine cDNAs from the *H-2 D^b* gene to retain murine CD8α interaction. Human *ß2m* was linked to the HLA N-terminus by a 15-amino acid glycine-serine peptide linker, followed by α1, α2, α3 chains, transmembrane and cytoplasmic domains with bovine growth hormone (bGH) poly-adenylation signal at the 3′-end. Phusion DNA polymerase combined HLA amplicons by overhang extension PCR. Six chimeric HLA alleles, HLA-*A*03:01*, *A*11:01*, *B*07:02*, *B*15:01*, *C*04:01*, and *C*07:02* were cloned into minimal mammalian expression vector, the pMA plasmid (GeneArt plasmid construction service, Thermo Fisher Scientific), as single haplotype sequentially using unique restriction enzyme sites to generate the AB*ab*-I transgene. For mouse genome targeting, AB*ab*-I transgene was cloned into hyperactive PiggyBac expression vector flanked by 5′- and 3′-ITRs using 5′-*Asc*I and 3′-*Pac*I restriction sites to generate the PiggyBac transgene targeting cassette. A hyperactive version of the PiggyBac transposase (hyPBase) was in vitro transcribed (*ivt*) and delivered as mRNA into zygotes. The six HLA PiggyBac transgene cassette DNA was mixed with the hyPBase (*ivt* mRNA) at a molar ratio of 2:1. Mixed DNA with *ivt* mRNA samples were pronuclei-microinjected into fertilized mouse oocytes of AB*ab*-A2[7] as described[36,37]. 10–12-week-old male AB*ab*-A2 mice were mating partners. 6–10-week-old female AB*ab*-A2 mice were super-ovulated before mating and zygotes were collected. NMRI mice served as pseudo-pregnant foster mothers. For HLA-I genomic confirmation in newborn AB*ab*-I mice, genotyping PCRs amplified all six HLA alleles using distinct allele-specific primers (Supplementary Table 2). Targeted locus amplification was performed by Cergentis B.V. for the determination of PiggyBac integration sites in the AB*ab*-I mouse genome.

### Immune phenotyping
HLA surface expression was analyzed in vitro in murine MCA205 cells using anti-human pan HLA-ABC (clone: W6/32, Biolegend) and anti-human ß2m antibodies (clone: TÜ99, Beckman Coulter). The ß2m antibody was selected for ex vivo blood analysis due to its superior staining of chimeric HLA fusion proteins, where ß2m is covalently linked to the HLA heavy chain, and it is specific to human ß2m without cross-reactivity to other species. *Blood preparation:* Cells in 50 μl blood per mouse were blocked for Fc receptors III/II using anti-CD16/32 antibody (Clone: 93, Biolegend). Red blood cells were lysed using erythrocytes lysing buffer (ACK buffer) before staining and flow cytometry. *Lymphoid organs preparation:* spleen and lymph nodes (axillary, brachial, mediastinal, inguinal, and mesenteric) from 8 to 12 weeks old mice were mashed and strained through a 70 μm cell strainer. ACK-lysed spleen and lymph node cells were suspended into single cells through a 40 μm strainer. *Staining for absolute T cell numbers:* samples were stained using antibodies specific for CD3 (Clone: 145-2C11), CD8 (Clone: 53-6.7), and CD4 (Clone: GK1.5) from Biolegend. 11 μl volume (11,880 beads) of CountBright counting beads (Thermo Fisher Scientific) per sample were used for T cell quantification by flow cytometry. A complete list of antibodies and their sources is provided in Supplementary Table 3.

### Genomic DNA extraction
Cells from lymphoid organs and whole blood from mice were prepared and pooled for CD3⁺ enrichment through bead-based magnetic separation using EasySep mouse T cell untouched isolation kit (Stemcell Technologies). Untouched CD3⁺ cell populations were surface stained with fluorescent antibodies specific for CD3, CD8, and CD4 markers, and the CD8⁺ cells were FACS sorted. Phenol-chloroform extraction was used to isolate genomic DNA from FACS-enriched CD3⁺CD8⁺ T cells (≥ 97% purity obtained for TCR repertoire sequencing; Supplementary Fig. 4E).

### TCR repertoire deep sequencing
Deep sequencing of TCRα and TCRß immune repertoire was performed at Adaptive Biotechnologies (Seattle). ImmunoSEQ platform precisely quantifies all V-J gene combinations in cell populations using a multiplex PCR-based assay with bias correction in the first step to minimize amplification bias[38,39]. The assay also provides quantitative abundance data with a sensitivity of 1 in 200,000 T cells. 2–4 μg of genomic DNA/sample obtained from $3 \times 10^5$ mouse (average in mice) CD8⁺ T cells were deep sequenced. ImmunoSEQ analyzer portal provided access to the deep sequenced TCR data. Computational and statistical analysis was performed and analyzed as described for CD4⁺ TCR repertoire using the R program[15]. We included all the reads with V genes identified as *TRBV20*, in contrast to our previous work[15], where only the *TRBV20-01* chains were counted. The counts of the TCR clones were normalized based on the rounds of initial PCR and it reflected the real abundance of the clones from the deep sequencing samples. We used the normalized counts to evaluate the true richness of the repertoires using the iChao1 estimator. The Jaccard index (J) measures the similarity of TCRα and TCRß clones between and within the AB*ab*-I and AB*ab*-A2 mouse groups. The J index score, calculated as $J(A, B) = (A \cap B)/(A \cup B)$, compares the total number of shared clones to the total number of unique clones, where A and B represent any two TCRα or TCRß repertoire samples. TCR sequences were designated based on Immunogenetics (IMGT) nomenclature[40].

### Retroviral TCR transduction of murine splenocytes
Platinum-E188 (Plat-E) ecotropic retrovirus packaging cells were transfected at 80% cell confluency using Lipofectamine with pMP71 retroviral plasmids (3 μg/construct) encoding T cell receptor genes (configuration: TCRß-mCß-P2A-TCRα-mCα). Mouse constant region genes were used in the transgenic TCRs to minimize mixed pairing with the endogenous TCRs of AB*ab*-A2 T cells. 3 ml of virus super-natants were harvested twice at different time points using a 0.45 μm membrane filter for infecting murine splenocytes. Splenocytes from AB*ab*-A2 mice were pre-activated with anti-CD3 (1 μg/ml) and anti-

CD28 (0.1 μg/ml) antibodies (Biolegend) in the presence of 40 IU/ml recombinant mouse IL-2 (Novartis) with cell concentrations adjusted to $2 \times 10^6$ cells/ml. Transductions were performed twice at 48- and 72-h post-transfection. Post-second transduction, transduced splenocytes were cultured under cell concentrations of $2 \times 10^6$ cells/ml and supplemented with 50 ng/ml murine IL-15 (Novartis). The transduction efficiency of AB*ab*-A2 splenocytes was analyzed by flow cytometry with anti-mouse TCR-Cß antibody (Clone: H57-597, Biolegend) and ranged between 33 and 47% of TCR-transduced CD8+ T cells.

### In vitro CD8+ T cell stimulation assay

Overnight (16 h) co-cultures were performed with $5 \times 10^4$ TCR-transduced effector T cells and $5 \times 10^4$ target cells derived from peripheral blood cells or cells from lymphoid organs of AB*ab*-I mice as antigen presenting cell (APC). These AB*ab*-I APCs were pulsed with peptides (1 μM/peptide, GenScript), or with phorbol myristate acetate (PMA) and ionomycin (1 μM each) as a positive control. As negative controls, TCR-untransduced T cells from AB*ab*-A2 mice pulsed with peptides and TCR-transduced T cells co-cultured with HLA-I cells without cognate peptides were used. In some cases, we included no peptide as negative controls, and in other cases, this control had already been published[21,23,24]. IFN-γ amounts in the culture supernatants were measured by enzyme-linked immunosorbent assay (ELISA, BD Biosciences).

### Peptide immunizations in mice

A 200 μl injection suspension containing 100 μg of peptides in PBS, 100 μl of incomplete Freund's adjuvant, and 50 μg CpG oligonucleotides were injected subcutaneously on both lateral sides of the hind legs (100 μl per side). 10–20-week-old mice (6–7 mice/group) were immunized with a 4-week interval between prime and boost. As standard control, AB*ab*-I mice were similarly immunized with 100 μg of MAGE-A1$_{278}$ KVLEYVIKV peptide[11]. 7–10 days after the last immunization, peripheral blood cells were ACK-lysed and cultured with 1 μM respective specific or unspecific peptide in the presence of a Golgi plug (Brefeldin A, BD Biosciences) for 5–6 h. Cells were fixed, intracellularly stained, and analyzed for IFN-γ (Clone: XMG1.2, Biolegend), CD3, and CD8 expression by flow cytometry.

### Statistical analysis

All analyses were performed using GraphPad Prism version 10.4.1. Data are presented as mean ± standard deviation (SD). One-way ANOVA was used to assess overall statistical differences in experiments involving multiple groups. Pairwise comparisons were performed using two-tailed unpaired Student's $t$-test. $P$ values less than 0.05 were considered statistically significant and are represented as: *$P < 0.05$; **$P < 0.01$; ***$P < 0.001$; ****$P < 0.0001$; ns, not significant. Exact $P$ values for all comparisons are provided in the Source Data file.

### Reporting summary

Further information on research design is available in the Nature Portfolio Reporting Summary linked to this article.

## Data availability

The TCRα and TCRβ repertoire deep sequencing data have been deposited for public access by Adaptive Biotechnologies in the ImmuneACCESS database and are available under the following DOI links: [TCRα: https://doi.org/10.21417/AD2025NC1; TCRβ: https://doi.org/10.21417/AD2025NC2]. Raw sequencing files can be analyzed, exported, and downloaded from the ImmuneACCESS site, as previously described in ref. 15. AB*ab*-I mice are exclusively licensed to T-knife Therapeutics, Inc. and housed at the Max Delbrück Center for Molecular Medicine in Berlin, Germany. Any non-commercial academic researcher may request access via a standard NDA/MTA agreement with T-knife Therapeutics. All data are included in the Supplementary Information or available from the authors, as are unique reagents used in this Article. The raw numbers for charts and graphs are available in the Source Data file whenever possible. Source data are provided with this paper.

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

## Acknowledgements

This study was supported by research funding from the European Union (ERC Advanced Grant 882963 Neo-T, T.B.). This study was partly supported by T-knife Therapeutics Inc. We thank our internal members, I. Gavvovidis, G. Willimsky, and T. Kammertöns, for their valuable input and discussions. We thank T. Schüler for his valuable input and discussions. We thank E. Kieback for her immense support and scientific input in completing the mouse characterization studies. We thank M. Rösch and A. Deick for their invaluable support in maintaining our animals at the MDC mouse facility. We thank L. Knackstedt for his contributions to the functional characterization and FACS sorting discussions. We thank S. Schlosser for her technical support during the initial characterization phase. We thank A. Leschke and C. Scholl for their technical expertise in performing zygote microinjections. Figure 1 Created in BioRender. Dhamodaran, A. (2025) https://BioRender.com/wii3mup.

## Author contributions

T.B. conceived the AB*ab*-I humanized TCR mouse concept, mirroring the human situation. T.B. supervised all research activities and acquired funding. A.D. conceptualized and performed the in silico computational screen to define the HLA haplotype (six HLAs) for AB*ab*-I mouse generation. A.D. designed the project strategy, planned and executed almost all (some in collaboration with other authors) in vitro, ex vivo, and in vivo experiments, analyzed data, and constructed figures. A.D. wrote the paper with input from T.B. X.C. made critical contributions to TCR repertoire data analysis and validation of V(D)J diversity in AB*ab*-I mice through computational and statistical analyses. N.F. made major contributions to method establishment for genotyping, immune phenotyping, and TCR deep sequencing. D.A. contributed to AB*ab*-I mouse line establishment and ex vivo functional characterization. E.D. coordinated TLA sequencing and deciphered chromosomal integration sites of the AB*ab*-I transgene cassette. R.K. made crucial contributions to mouse model generation and supervised all zygote microinjection activities at the transgenic core facility. T.B. made critical contributions to data interpretation, discussion, and manuscript revision. A.D. supervised and coordinated all internal (flow cytometry, transgenic core facility, and animal facility) and external industry (Adaptive Biotechnologies and Cergentis) MTA/NDA discussions and collaborations.

## Funding

## Competing interests

T.B. and A.D. are inventors on a pending patent (PCT Nr. WO2023144087A1) jointly applied by the Max-Delbrück Center and T-knife Therapeutics with a title 'A non-human mammal comprising in its genome at least two human leukocyte antigen (HLA) class I alleles, methods of making such mammal and uses thereof' describing the mouse as a method tool for novel epitopes and TCR discovery. T.B. and A.D. is inventor on a pending patent (PCT Nr. WO2023139257A1) jointly applied by the Max-Delbrück Center and T-knife Therapeutics describing the KRAS-restricted TCR used in this study. A.D. and D.A. are former employees, and N.F. and E.D. are current employees of T-knife Therapeutics; they may hold stock and/or stock options in the company. T.B. is a founder and SAB (scientific advisory board) member of T-knife Therapeutics. The remaining authors declare no competing interests.
