## [Transparent Peer Review file · Nature Communications]

Mice with a diverse human T cell receptor repertoire selected on multiple HLA class I molecules.

Corresponding Author: Professor Thomas Blankenstein

Version 0:

Reviewer comments:

Reviewer #1

(Remarks to the Author)

The paper by Dhamodaran et al describes the generation and evaluation of new transgenic mouse line expressing 7 human HLA alleles on a background of complete human TCR α and β gene loci deficient for mouse TCR and MHC I. Authors previously generated ABab-A2 mice (previously named ABabDII, ref 7) and added six chimeric alleles in one tandem construct using PiggyBac transposon. Thorough analysis was performed comparing the novel ABab-I mice with ABab-A2. The introduced alleles frequently occur in the human population. Chimeric single chain HLA-b2m molecules with human $\alpha 1$ and $\alpha 2$ subunits and mouse $\alpha 3$ enable mouse CD8 docking. They observed more peripheral CD8 T cells, 4-fold broader TCR repertoire, longer CDR3 regions and T cell responses to viral and tumor antigens. This mouse line constitutes an valuable source to isolate high affinity TCRs after immunization in the absence of tolerance for human antigens for immunotherapy exploitations.

Comments:

1. The title states that the new mouse line contains a HLA haplotype. In the strict sense, this is true because all alleles were cloned in tandem. However a human HLA haplotype only contains three alleles. It might be worthwhile to state in the title that the mice express seven HLA alleles.
2. The TCR repertoire of the mouse is not sculpted by central tolerance to human antigens, exploiting also high affinity receptors. This is ideal for real foreign antigens like viral or cancer neoantigens, but might cause adverse events with respect to cancer associated antigens. This should be included in the discussion section.
3. Row 151-161 is frank on the instability of the mouse line over 5 generations. Mice were screened and high expressors were intercrossed. How stable is the line now? How many generations is the line propagated without loss of functionality? Please provide info.
4. Fig 2A-B, the upper dotplot row shows an CD4 CD8 flow cytometry plots of the mouse thymus. This does not look like a classical thymocyte plot ('swallow plot'). Is this due to the limited number of events. Show more events then. In addition, show % of DN, DP and SPs of multiple mice in bargraphs.
5. Row 113, statement on CD44+ T cells is expected to be followed by reference, not a figure number.
6. Fig 6: add statistics
7. Fig 9: were B6 organs included as control APC?

Reviewer #2

(Remarks to the Author)

The work by Dhamodaran et al. is important and a significant advance towards generating mice with human T cell repertoires for potential clinical translation. The manuscript is well written, and the immunologic analyses and results seem solid.

There are a few issues that need clarification/correction with regards to how the mice were generated:

- 1) The authors state that "allele-specific PCRs confirmed the presence of the six HLA alleles in the founder animals." This data should be presented, at least as a supplemental figure. Genomic presence of the full six alleles and full transposon needs confirmation.
- 2) The authors state, "The PiggyBac transposon technology proved to be efficient but bears the risk of losing copy numbers of multiple chromosomal integration sites." It seems more likely that the number of integrations in the animals decreases over time with breeding because they are bred out, not that the transposons hop out of the genome. As the integrations are on different chromosomes, they are likely lost through breeding. This sentence needs clarification.
- 3) The version of the hyperactive piggyBac transposase should be specified.
- 4) The size of the transposon in figure 1 should be specified.
- 5) Figure 1 incorrectly depicts piggyBac integration. Below the oocyte transcription sites, they have depicted two TTAA sites. piggyBac does not integrate into 2 TTAA sites. piggyBac integrates into one TTAA site and produces target site duplication (two TTAA sites, one on each end of the transposon). This figure needs to be corrected.

Reviewer #3

(Remarks to the Author)

In this manuscript entitled "Mice with a diverse human T cell receptor repertoire selected on a complete HLA- class I haplotype, Dhamodaran et al generated mice that are deficient in murine TCRs and MHC-I but have a full HLA-I haplotype and full TCRab genes. These HLA-I molecules could present peptides to T cells and the resulting CD8+ T cells were able to recognise and respond to viral and tumour-specific antigens.

The authors suggest this murine model will be a powerful tool for epitope and TCR identification which they suggest may be beneficial for ATT treatment in the future for human. The mouse system used here has a chimeric construct of HLA, with a murine alpha 3 domain of the heavy chain and this will impact CD8 cofactor binding, therefore there will be some serious difference in thymic selection and activation in human that could make the results not replicable in human setting and especially in a clinical context.

This is an interesting mouse model which may indeed be useful to many in the future. While there is some nice data shown, several of the conclusions are significantly overstated and not supported by the results or the data.

Comments and questions

While this new model will be useful for epitope and TCR identification it is not clear how TCRs isolated from mice are any more useful than soluble expressed TCRs for grafting and ATT therapy. Could the authors please elaborate on this? In addition, the presence of murine CD4+ T cells on the development and selection of CD8+ T cells is unclear, and whether those CD8+ T cells will be similarly present in human, and tolerant?

The overall conclusion that this model will allow large number of patients to receive ATT is quite strong and should perhaps be toned down, or further supported by data.

Is there any evidence that all these HLA-I molecules are expressed at the cell surface?

CD8+ T cell activation requires the help of CD4 T cells which have been raised on murine MHC-II in this system. Do the authors think that has any implications?

Are ABab mice (without any HLA-I insertion) able to survive? This might have been a nice control group.

To compare the CD8+ T cell number and TCR diversity, it would have been a good control to include HLA-A2 in the mice with the 6 HLA-I molecules for a more direct comparison.

Are the mice on a C57/B6 backbone? Is this why these mice are used as a control? If so please state this in the intro, methods and results.

The rationale for the use of "α3, transmembrane and cytoplasmic domains from murine H-2 Db gene and fused to the human 2m gene" is unclear? Why not keeping mouse b2m as well?

Please hyphenate HLA-I

Italic should be used for gene and alleles

human MHC is HLA so please use HLA

Figure 1B, it is not clear what the stats bars are referring to (which groups have been compared)

Figure 2C, why are there so many cells squashed against the Y axis? How do you differentiate CD44+ vs CD44high expression?

Line 116 "which is likely caused by a partial mismatch between mouse and human TCR-MHC interactions" do you mean within the same mouse? If so, are the native murine TCR and MHC not entirely not knocked out?

Line 122 “showed that multiple HLA-I alleles selected a diverse repertoire that rescued...” since no TCR repertoire data is shown this claim is not substantiated. At most one can comment on the number, proportion and phenotype from the data shown.

Line 132 “These data indicate efficient expression of multiple HLA alleles” I am surprised that only 40% of cells are staining for HLA-I since all nucleated cells should have HLA-I expression. Further, it is impossible to tell how many of the HLA-I molecules are being expressed at the cell surface using a b2m antibody and all references of “multiple HLA molecules being expressed should be removed”

Fig 3 data do not show the claim of “their possibilities to trigger CD8+ T cell responses against peptide antigens.” It only show expression of HLA molecules on cell surface.

Fig 4 A, what re the lines in the data in the top left quadrant representative of?

There is a lot of rationale/interpretation/discussion throughout the results which is unusual.

The results related to Figure 5, please include in the text how the experiments were done, ie what method was used and what organs were assessed, and from how many animal?

Given there were different numbers of CD8s in the periphery of the ABab-A2 vs ABab-I mice, how were the TCR data normalised?

Fig 6 results suggest that there is a large overlap between the CD8+ TCR repertoire of ABab-A2 and ABab-I mice which is extremely surprising given that they have completely different HLA class I, this does not make sense, and it look like the sequencing data is coming from 1 animal only (to confirm) so perhaps some issue with the analysis or result here?

Line 172 “increased smaller” please reword

Fig 7, is the trend the same if looking at # vs %? The “longer CDR3” for the sub heading refereeing to fig 7 is based on 0.5 nucleotide difference between the 2 mice group, here again if the result is coming from 1 animal the conclusion are not supported.

The results would benefit from 1-2 sentences explaining the rationale of what was done followed by the method then results.

Do not alternate between HLA-I and MHC-I throughout

Page 10 “All six HLA I alleles present peptides ex vivo” this title is incorrect as allele do not present peptide.

Page 11 “T cells for HLA allele recognition bound with its respective model epitope” T cells don’t recognize allele, and what is a “model” epitope?

Fig 9, I can’t see any no-peptide negative control for any of the cell lines? Were the isolated cells used as APCs for the cell lines? Are the T cell lines HLA-I null? How do you know there is no self-presentation? A control of peptide + transduced T cells (with no ABab cells) is needed to ensure that there is no auto-activation.

In the discussion, it is not clear how these mice will enable epitope-specific TCR discovery any easier than scRNAseq directly from human cells and as such the discussion about TAA requires more information for the readers to understand the link.

The discussion is the first time readers are informed that the HLA-I were transferred into the HLA-A2 expressing mice so they have 7x HLA-I molecules but at other times results were referred to as “non-A2”

It is not clear how the discussion about HLA/TCR evolution contributes much to the story. Indeed, a lot of the discussion reads very different to the intro and how the results story was set up.

Page 12 “ABab-I mice being a useful tool for epitope and TCR discovery” how would the mouse model described here been a better tool for epitope discovery compared to the use of human PBMCs?

How did the authors compare pre-selection vs post-selection as per the conclusions?

The methods would benefit from slightly more details.

Reviewer #4

(Remarks to the Author)

Version 1:

Reviewer comments:

Reviewer #1

(Remarks to the Author)

I am completely satisfied with the updated version of this manuscript.

Reviewer #2

(Remarks to the Author)

The authors have appropriately responded to my comments.

Reviewer #3

(Remarks to the Author)

The authors have significantly enhanced the manuscript, but there are still some points to clarify or address with details to help below.

Now the flow of the story is clearer. The first paragraph of the introduction could be expanded to make it easy to follow for those outside of the field. There are huge jumps, assuming the reader can fill in the knowledge gaps. I.e. "TAA specific HLA-I TCRs can preferentially be isolated from transgenic mice, because humans are typically tolerant to TAA and contain usually only low-avidity T cells". This could be expanded on, to explain that TAA are often self-peptides, and as such during selection high avidity T cells are deleted, leaving only the low avidity ones. As such, these low avidity ones are not optimal in mounting an immune response. To counteract this, this new model ...

Line 46 "Transgenic mice are suitable to isolate TCRs from a naïve repertoire through immunization that triggers T cell memory responses" please rephrase, because immunizing will activate T cells (as you say), so you will not be isolating naïve cells (as per the first part of the sentence)...

Line 60, please add "additional" into the sentence, so as to remind the readers these mice already have A2 expression and now have 6 additional HLA-I molecules.

Fig 3, why did the authors select B2m rather than staining with the commonly used pan-HLA-I antibody clone W6/32? In addition, the reply to question #15 states "One explanation could be the folding characteristics of full human HLAs vs chimeric molecules, which could hinder epitope accessibility by the anti-human B2m antibody." This is not correct and was previously shown, also here again w6/32 would have been a better Ab to use. Even if the authors are "are confident that all six HLA-I molecules are functional and expressed" there is no data to demonstrate this presented here.

Figure 9, there is still no reference to negative controls (TCR transduced cells + HLA-I cells with NO PEPTIDE) in either the text, figure or figure legend. This needs to be amended. This control (which you have done according to the rebuttal) is critical to show the transduced TCR clones are not non-specifically responding to the HLA-I expressing cells. It would be good to show some raw data (FACS Plots) either in the main figure or in supplemental data.

Please start the discussion with a recap of the rationale and why this is being done. I.e. we want to generate new tools to be able to identify TCRs that could be used for therapy as human derived ones are low avidity.

It is now much clearer the role of using these murine derived TCRs for ATT as they may be of higher avidity than TCRs found in humans (however this remains an hypothesis), and the addition of the discussion about off target effects is interesting. However, the authors discuss ways they have tried to reduce off target effects (i.e. alanine scan) but it would be good to broaden this to how these would need to be tested by anyone before being used for therapeutics.

I feel that the flow of the discussion is a bit disjointed. Consider reorganising to enhance readability, a suggested flow below

- Rationale for why you want to make these tools
- How you made these tools
- Why you selected these HLAs and protocols
- Then talk about the TCRs, the caveats around tolerance and testing cross-reactivity (currently this is quite up front, before the discussion on the model itself)
- Finally, how else these mice could be useful

Version 3:

Reviewer comments:

Reviewer #3

(Remarks to the Author)

The authors have answered my questions, provided a detailed rebuttal and have made their manuscript clearer. I have no further questions, just some corrections to incorrect comments regarding W6/32 Ab use and binding from the rebuttal.

Minor comments (about the rebuttal only)

Line 138 – The anti-human pan HLA-ABC, is it pan HLA class I, even for non-classical HLA-I or only for classical HLA-I? W6/32 is a pan HLA class I molecules, not only ABC.

“The W6/32 antibody stains all HLA-I molecules indirectly by targeting residues in the N- terminus of the human β 2-microglobulin (β 2m) molecule.”

This is incorrect, the W6/32 Ab bind the HLA-I molecule itself, and the β 2m, making it specific to HLA class I, both classical and non-classical, but not to HLA-like molecule such as CD1 or MR1 molecules. The crystal structure of the W6/32-HLA-I complex is showing this with great details (PDB code 7T0L).

“As a result, it does not allow for the precise identification of which HLA-I molecules are expressed on the cell surface”

This is the case for all pan-Ab, while it does not allow for the precise identification of which HLA-I is expressed, it does show that there is an HLA-I molecule expressed.

“Therefore, whether one uses the β 2m antibody or the W6/32 antibody to stain HLA-I makes little difference, as both methods are indirect and non-specific for any of the six HLA-I molecules.”

β 2m can be associated to MHC-like molecule such as CD1 or MR1, so staining only with β 2m Ab is a real issue, especially for this study.

“Additionally, the referenced article, which maps the epitope for W6/32, demonstrates that W6/32 does not react with recombinant human β 2m” This is incorrect. All the work using recombinant human β 2m show that the w6/32, used for example in Surface plasmon resonance assay” bind the human β 2m. The structure of W6/32 with HLA-I (PDB code 7T0L) is in complex with recombinant human β 2m, clearly showing binding.

“This could explain why W6/32 did not stain well in our experiments.”

This comment is very confusing, as this is not what Supp Fig 2 shows, the w6/32 staining is comparable to the β 2m-Ab staining.

We would like to thank the Reviewers for their comments and constructive criticism. All changes in the main manuscript and supplementary material files are color highlighted in 'yellow'. We want to draw the reviewers' attention to the fact that the following figures have been revised or newly added: Figure 1, 2, 4, 6, and Supplementary Figure 1 and Supplementary Table 2.

The revised manuscript (uploaded as a PDF) has new line numbers due to the addition of sentences throughout. We have addressed all reviewers' comments by referencing the updated line and page numbers.

Reviewer #1

The paper by Dhamodaran et al describes the generation and evaluation of new transgenic mouse line expressing 7 human HLA alleles on a background of complete human TCR α and β gene loci deficient for mouse TCR and MHC I. Authors previously generated ABab-A2 mice (previously named ABabDII, ref 7) and added six chimeric alleles in one tandem construct using PiggyBac transposon. Thorough analysis was performed comparing the novel ABab-I mice with ABab-A2. The introduced alleles frequently occur in the human population. Chimeric single chain HLA-b2m molecules with human a1 and a2 subunits and mouse a3 enable mouse CD8 docking. They observed more peripheral CD8 T cells, 4-fold broader TCR repertoire, longer CDR3 regions and T cell responses to viral and tumor antigens. This mouse line constitutes a valuable source to isolate high affinity TCRs after immunization in the absence of tolerance for human antigens for immunotherapy exploitations.

Comments to Reviewer #1:

1. The title states that the new mouse line contains a HLA haplotype. In the strict sense, this is true because all alleles were cloned in tandem. However a human HLA haplotype only contains three alleles. It might be worthwhile to state in the title that the mice express seven HLA alleles.

We changed the title on page 1 to: "Mice with a diverse human T cell receptor repertoire selected on multiple HLA class I molecules".

2. The TCR repertoire of the mouse is not sculpted by central tolerance to human antigens, exploiting also high affinity receptors. This is ideal for real foreign antigens like viral or cancer neoantigens, but might cause adverse events with respect to cancer associated antigens. This should be included in the discussion section.

On page 12, in line 292, we added the below sentences after ... 'humans are typically tolerant and bear only low-affinity TCRs':

"TCRs specific for TAA bear the risk of exhibiting off-target reactivity and toxicity, because these TCRs were not negatively selected against human antigens. Due to the protein sequence discrepancy between human and mice, there may be potential cross-reactivity to human self-antigens. Therefore, we previously employed a series of experiments with the aim to minimize the risk of off-target toxicity. These experiments included alanine scan to determine the TCR contact residues of the peptide, searching the human proteome for peptides with the same TCR contact motif and testing these peptides for recognition by the TCR under investigation. Furthermore, we tested the TCRs for recognition of a library of self-peptides and, for testing

HLA alloreactivity, a library of human LCL cell lines covering a large number of MHC-I and MHC-II alleles (Ref. 11, 12).”

3. Row 151-161 is frank on the instability of the mouse line over 5 generations. Mice were screened and high expressors were intercrossed. How stable is the line now? How many generations is the line propagated without loss of functionality? Please provide info.

On page 13, in line 312, we added the below sentence after “... ‘both, the thymus and peripheral lymphoid organs’:

“Since generation F8, the AB*ab*-I line was stable and yielded uniformly high CD8⁺ cell numbers in the periphery. Afterwards, we have propagated the line for more than 10 generations with continuously high and stable peripheral CD8⁺ cell numbers.”

4. Fig 2A-B, the upper dotplot row shows an CD4 CD8 flow cytometry plots of the mouse thymus. This does not look like a classical thymocyte plot (‘swallow plot’). Is this due to the limited number of events. Show more events then. In addition, show % of DN, DP and SPs of multiple mice in bargraphs.

Thanks to Reviewer #1 for his comments on our thymocyte analysis. We modified Fig 2A-B, the upper dotplot row showing more events now for both top and bottom analyses. In addition, the gating has been changed to show the view of so-called classical ‘Swallow plot’ view. In addition, we show % of DN, DP and SPs of multiple mice as Fig. 2C, and changed the complete Fig. 2 accordingly. Please see the new Fig. 2 in the manuscript file.

5. Row 113, statement on CD44⁺ T cells is expected to be followed by reference, not a figure number.

Thanks for pointing this out. We have now included the correct reference on page 6, line 118, and on page 23, line 572 (Almeida *et al.*, J Exp. Med. 2001), that was originally supposed to be cited.

6. Fig 6: add statistics

As mentioned on page 9, lines 200-201, there is no significant difference among any preferred usage of V_α/V_β in the preselection pool between AB*ab*-A2 and AB*ab*-I mice which was to be expected. Additional statistical analyses on the complex preselection pool data set would not make sense. However, differences occur in the post-selection pool. Therefore, we performed statistical analyses on highlightable V_α/V_β chains mentioned in ‘lines 205-208’. Please, see the new Fig. 6 in the manuscript file.

7. Fig 9: were B6 organs included as control APC?

We did not include B6 organs in this experiment, because the human peptides cannot be presented by the mouse MHC-I molecules and the human TCRs used are not restricted to the mouse MHC-I molecules. Instead, we performed a negative control in the co-culture reaction shown in Figure 9 by including Untransduced T cells. It is important to point out that it was known before that the chosen peptides are presented by the respective HLA I molecule and that the used TCRs were restricted to the respective HLA I molecule and specific for the respective peptide (Ref. 21-25), as shown in Figure 9A.

Reviewer #2

The work by Dhamodaran et al. is important and a significant advance towards generating mice with human T cell repertoires for potential clinical translation. The manuscript is well written, and the immunologic analyses and results seem solid.

There are a few issues that need clarification/correction with regards to how the mice were generated:

Comments to Reviewer #2:

1) The authors state that “allele-specific PCRs confirmed the presence of the six HLA alleles in the founder animals.” This data should be presented, at least as a supplemental figure. Genomic presence of the full six alleles and full transposon needs confirmation.

Thanks to Reviewer #2 for the comment. We now present allele-specific PCR confirmation using electrophoresis gel image as a new Supplementary Fig. 1. This confirms the genomic presence of the full six alleles. This we cite this as the new ‘Supplementary Fig. 1’ on page 5, lines 95-97, at the end of the sentence... ‘After pronuclei injection, transgenic mice were generated and allele-specific PCRs confirmed the presence of the six HLA alleles in the founder animals’.

Additionally, we added sentences on Page 17, lines 427-430, after... ‘NMRI mice served as pseudo-pregnant foster mothers’:

For HLA-I genomic confirmation in newborn ABab-I mice, genotyping PCRs amplified all six HLA alleles using distinct allele-specific primers (Supplementary Table 2). Targeted locus amplification was performed by Cergentis B.V. for the determination of PiggyBac integration sites in the ABab-I mouse genome.

Transposon confirmation:

The genomic presence of transposons with insertion sites is shown in Supplementary Figure 3 (old Supplementary Figure 2) using targeted locus amplification (TLA) analysis, where primers amplify six-HLA cassette mapping the ITR insertion sites.

2) The authors state, “The PiggyBac transposon technology proved to be efficient but bears the risk of losing copy numbers of multiple chromosomal integration sites.” It seems more likely that the number of integrations in the animals decreases over time with breeding because they are bred out, not that the transposons hop out of the genome. As the integrations are on different chromosomes, they are likely lost through breeding. This sentence needs clarification.

We agree with Reviewer #2’s comments regarding the sentence and have revised it for clarity, as we provided the PiggyBac transposase enzyme as mRNA. On page 12, in lines 306-309, we have corrected the sentence to:

‘The PiggyBac transposon technology proved to be efficient but bears the risk of losing copy numbers of multiple chromosomal integration sites, likely due to the gradual loss of integrations over successive breeding generations rather than transposon excision’.

3) The version of the hyperactive piggyBac transposase should be specified.

First, we now specified this in the ‘Materials and Methods’ section, in lines 420-423: We edited the sentence, in line 420, starting... ‘A hyperactive’:

‘A hyperactive version of the PiggyBac transposase (hyPBase) was *in vitro* transcribed (*ivt*) and delivered as mRNA into zygotes. The six HLA PiggyBac transgene cassette DNA was mixed with the hyPBase (*ivt* mRNA) at a molar ratio of 2:1.’

Second, we changed Figure 1 accordingly to highlight the use of ‘hyperactive PiggyBac transposase (hyPBase)’.

4) The size of the transposon in figure 1 should be specified.

We now mention the size of transposon in Figure 1.

5) Figure 1 incorrectly depicts piggyBac integration. Below the oocyte transcription sites, they have depicted two TTAA sites. piggyBac does not integrate into 2 TTAA sites. piggyBac integrates into one TTAA site and produces target site duplication (two TTAA sites, one on each end of the transposon). This figure needs to be corrected.

We thank Reviewer #2 for pointing out this error. We changed Figure 1. Please see the new Fig. 1 in the manuscript file on page 28.

Reviewer #3

In this manuscript entitled “Mice with a diverse human T cell receptor repertoire selected on a complete HLA- class I haplotype, Dhamodaran et al generated mice that are deficient in murine TCRs and MHC-I but have a full HLA-I haplotype and full TCRab genes. These HLA-I molecules could present peptides to T cells and the resulting CD8+ T cells were able to recognise and respond to viral and tumour-specific antigens.

The authors suggest this murine model will be a powerful tool for epitope and TCR identification which they suggest may be beneficial for ATT treatment in the future for human. The mouse system used here has a chimeric construct of HLA, with a murine alpha 3 domain of the heavy chain and this will impact CD8 cofactor binding, therefore there will be some serious difference in thymic selection and activation in human that could make the results not replicable in human setting and especially in a clinical context.

This is an interesting mouse model which may indeed be useful to many in the future. While there is some nice data shown, several of the conclusions are significantly overstated and not supported by the results or the data.

Comments to Reviewer #3:

Comments and questions

1) While this new model will be useful for epitope and TCR identification it is not clear how TCRs isolated from mice are any more useful than soluble expressed TCRs for grafting and ATT therapy. Could the authors please elaborate on this?

On page 3, lines 44-46, we had written: “TAA-specific HLA-I TCRs can preferentially be isolated from transgenic mice, because humans are typically tolerant to TAA and contain usually only low-avidity T cells.” In other words, mice are not tolerant to many human tumor antigens/epitopes, if the sequence of mouse and human differs from each other. Therefore, higher affinity TCRs can be isolated from antigen-negative mice (Ref. 11).

We have to admit, however, that we don't understand what the Reviewer means with “soluble expressed TCRs for grafting and ATT therapy”.

2) In addition, the presence of murine CD4+ T cells on the development and selection of CD8+ T cells is unclear, and whether those CD8+ T cells will be similarly present in human, and tolerant?

On page 6, lines 119-124, we had written: “In splenocytes, we observed no significant change in CD44 expression among CD4+ T cells in both ABab-I and ABab-A2 mice (old figure Fig. 2, C and D) [now corrected to (Fig. 2, D and E) in the revised manuscript]. However, CD44 expression by CD4+ T cells of both mouse lines was increased in comparison to control mice, which is likely caused by a partial mismatch between mouse and human TCR-MHC interaction and impaired CD4+ T cell development, since the human TCRs in ABab-A2 and ABab-I mice are selected by mouse MHC-II molecules (Ref. 15).”

Otherwise, the CD4 T cells appear normal but we also showed that the CD4/CD8 ratio substantially shifts, depending on whether the mice have one or multiple HLA-I genes.

Therefore, on page 7, lines 141-144, we had written: “Within CD3+ peripheral blood cells, there were 13% CD8+ T cells in ABab-A2, 45% in ABab-I and 41% in C57BL6/N control mice (Fig. 4A). CD4+ T cells were 63% in ABab-A2, 41% in ABab-I and 53% in control mice (Fig. 4A).”

The question whether CD8⁺ T cells will be similarly present in human, and tolerant, is interesting. The answer is yes and no and depends on the expression of the antigen in the thymus. MAGE-A1 is expressed in the thymus (in mTECs) and only low-affinity TCR can be isolated from humans. In mice for which MAGE-A1 is foreign, high-affinity TCR can be generated (Ref. 11). In contrast, the TCR specific for Melan-A/MART-1 can be identical between mice and humans, at least the alpha-chain, which is mainly responsible for specificity in this case. The reason is that the epitope is not expressed in the human thymus, thus no central tolerance of MART-1 exists in humans (DOI: 10.1002/eji.201444499).

3) The overall conclusion that this model will allow large number of patients to receive ATT is quite strong and should perhaps be toned down, or further supported by data.

We agree with this critique, as “large” is not defined.

In the last sentence of the abstract, lines 34-35, we wrote: “... which could increase the number of cancer patients amenable to TCR-T treatments.”

We now changed on page 4, line 73 to: “... allowing, in the future, cancer patients beyond those expressing HLA-A2 to benefit from ATT treatments.”

4) Is there any evidence that all these HLA-I molecules are expressed at the cell surface?

The Reviewer raises an interesting point. Remarkably, there are no specific antibodies for almost all of the HLA-I molecules. We tried some but they were not specific. So, we needed to stain with the anti- β 2m antibody. Flowcytometry with this antibody showed much stronger expression for the multiple compared to the single HLA-I mice and all six HLA-I genes were driven by the same promoter. Two lines of experiments made us confident that all six novel HLA-I molecules are cell-surface expressed. In Figure 9, it is shown that spleen cells from ABab-I mice can present peptides to T cells that are specific for a known peptide and restricted to one of the six HLA-I molecules. In Figure 10, we show that immunization with these six peptides, each restricted by only one of the six HLA-I molecules, can induce and expand specific CD8 T cells. We think, these data leave little doubts that all six HLA-I molecules are functionally expressed.

5) CD8⁺ T cell activation requires the help of CD4 T cells which have been raised on murine MHC-II in this system. Do the authors think that has any implications?

On page 19, lines 496-497 we wrote that immunization was done with a suspension containing CpG oligonucleotides in addition to the peptide. CpG circumvents CD4 T cell help. Therefore, we think the mouse MHC-II has no implication.

6) Are ABab mice (without any HLA-I insertion) able to survive? This might have been a nice control group.

Yes, ABab mice without HLA-I genes are vital and have been described in Ref. 7. They have mouse MHC-I genes, which render them useless for TCR generation and we did not continue to breed them. We do not have a mouse line with human TCR Alpha-Beta gene loci and no mouse or human MHC-I genes. These mice probably would not have CD8 T cells or only tiny amounts, which are not MHC-I-restricted. Under SPF condition, such mice likely would be vital.

7) To compare the CD8+ T cell number and TCR diversity, it would have been a good control to include HLA-A2 in the mice with the 6 HLA-I molecules for a more direct comparison.

This is what we did. We compared ABab-I mice with seven HLA-I molecules (six novel and HLA-A2) to ABab-A2 mice with just HLA-A2. We had made this clear in the first result paragraph on page 5, lines 90-91 by saying: “The PB ITR-flanked targeting vector carrying six HLA genes was co-injected with the hyperactive PB transposase mRNA into fertilized ABab-A2 mouse oocytes.”

8) Are the mice on a C57/B6 backbone? Is this why these mice are used as a control? If so please state this in the intro, methods and results.

No, the mice are a mixture of 129SV, C57Bl/6 and BALB/c. We were concerned that the ABab-A2 mice with seven gene-modified gene loci were poorly breeding. Therefore, we kept them initially outbred. However, meanwhile they may be inbred due to extensive inbreeding. We thought, therefore, C57BL/6 mice, the gold standard for mouse immunology was the best control.

9) The rationale for the use of “ α 3, transmembrane and cytoplasmic domains from murine H-2 Db gene and fused to the human β 2m gene” is unclear? Why not keeping mouse β 2m as well?

The mouse α 3 domain was important to allow CD8 binding. Transmembrane and cytoplasmic regions were of mouse origin, because they were not involved in TCR interaction, like the α 3 domain. We used human β 2m, because we were afraid that mouse β 2m would not appropriately bind to HLA-I molecules. We fused it to the HLA-I genes to ensure equimolar amounts.

10) Please hyphenate HLA-I, Italic should be used for gene and alleles, human MHC is HLA so please use HLA

Changes have been made accordingly.

11) Figure 1B, it is not clear what the stats bars are referring to (which groups have been compared)

We believe Reviewer #3 is referring to Figure 2B, not 1B, as there are no A and B panels in Figure 1.

We did notice that the stats bars did not appear in the earlier PDF version of the manuscript file (544059_0_art_file_9646472_sk9r6j). It could be technical error (word to PDF conversion!) that Reviewer #3 cannot clearly visualize stats bars. As we do not see the stats bars representing *P values (stars)* in the PDF version of the manuscript file on Figures 2, 3 and 4.

As stated in lines 105-115, on page 5-6, and in lines 734-736, on page 30, the statistical bars in this bar graph compared ABab-A2 mice (n=5) with ABab-I mice (n=5). However, to improve visual clarity, we have now completely revised Figure 2, with precise statistical bars placed directly above the compared groups and shortened error bar widths.

12) Figure 2C, why are there so many cells squashed against the Y axis? How do you differentiate CD44⁺ vs CD44^{high} expression?

We understand Reviewer #3's concerns. We changed Figure 2 entirely. So, Figure 2C is now Figure 2D. We moved the events (cells) further along the x-axis in Figure 2D. We believe now we can nicely distinguish between CD44⁺ vs CD44^{high} expressing populations. Please see the new Fig. 2 in the manuscript file.

13) Line 116 "which is likely caused by a partial mismatch between mouse and human TCR-MHC interactions" do you mean within the same mouse? If so, are the native murine TCR and MHC not entirely not knocked out?

No, not within the same mouse. Murine TCR alpha and beta and murine MHC-I (β 2m and H2-D^b) are completely knocked-out. The sentence refers to coevolution of MHC and TCR gene loci. Previously, we had shown in mice that the human TCR repertoire is well generated by human HLA molecules but less efficient by mouse MHC molecules (Ref. 15). This was done in CD4 T cells, since the mice contained either a single mouse or a single human HLA-II gene.

14) Line 122 "showed that multiple HLA-I alleles selected a diverse repertoire that rescued..." since no TCR repertoire data is shown this claim is not substantiated. At most one can comment on the number, proportion and phenotype from the data shown.

We agree and changed the sentence on page 6, lines 127-128 (new line numbers) to: "... showed that positive thymic selection by multiple HLA-I molecules was more effective and rescued..."

15) Line 132 "These data indicate efficient expression of multiple HLA alleles" I am surprised that only 40% of cells are staining for HLA-I since all nucleated cells should have HLA-I expression. Further, it is impossible to tell how many of the HLA-I molecules are being expressed at the cell surface using a β 2m antibody and all references of "multiple HLA molecules being expressed should be removed".

We agree with the Reviewer that 40% of the cells staining with the anti-human β 2m antibody is low. We believe that this antibody less efficiently stains β 2m when fused to HLA-I molecules. One explanation could be the folding characteristics of full human HLAs vs chimeric molecules, which could hinder epitope accessibility by the anti-human β 2m antibody. However, as mentioned above, we are confident that all six HLA-I molecules are functional and expressed, because spleen cells of AB*Ab*-I mice can present peptides specific for each one of the six HLA-I molecules to TCR-transduced T cells specific for the respective peptide and stimulate their IFN- γ production (Figure 9). Additionally, immunization of AB*Ab*-I mice with peptides known to be presented by each one of the six HLA-I molecules stimulates activation and expansion of peptide-specific CD8⁺ T cells (Figure 10).

Agreeing with the Reviewer, we removed..."efficient expression of multiple HLA molecules being expressed'...

We now changed the sentence on page 7, lines 137-138 to...:

'These β 2m staining data suggest HLA expression on the surface of AB*Ab*-I lymphocytes.'

16) Fig 3 data do not show the claim of "their possibilities to trigger CD8⁺ T cell responses against peptide antigens." It only shows expression of HLA molecules on cell surface.

We agree. Sentence on page 6-7, old lines 132-134 has been deleted and replaced with...:

'These β 2m staining data suggest HLA expression on the surface of ABab-I lymphocytes.' On page 7, lines 137-138.

17) Fig 4 A, what re the lines in the data in the top left quadrant representative of?

Thanks for pointing this out. We figured out that it is the low resolution which causes the broken lines effect in the dotplot. This was an error from our FlowJo analysis. We now corrected the dot plot with improved resolution of the events with low signal intensity. Please see the corrected Fig. 4A in the manuscript file.

18) There is a lot of rationale/interpretation/discussion throughout the results which is unusual.

We carefully analyzed all nine result sections for content regarding rationale/ interpretation/discussion. We could not identify a lot of rationale/ interpretation/ discussion, therefore it would have been helpful, if the reviewer had specifically pointed to paragraphs which appeared unusual. The critique is also difficult to address, because under point 24 the reviewer writes: "The results would benefit from 1-2 sentences explaining the rationale ...".

19) The results related to Figure 5, please include in the text how the experiments were done, i.e. what method was used and what organs were assessed, and from how many animals?

On page 8, lines 167-172, we added: "The TCR repertoire was determined by a multiplex PCR. Therefore, genomic DNA of purified CD8+ T cells from pooled spleen and lymph nodes was isolated. The genomic DNA equivalent of 3×10^5 (average in mice) CD8+ T cells per mouse was deep sequenced. Five mice per group were analyzed. Computational analysis was performed by ImmunoSEQ™ analyzer, and for the identification of specific V, D, and J genes, the International Immunogenetics (IMGT®) nomenclature was used."

We had indicated in the legend of Figure 5, that five mice per group were analyzed.

20) Given there were different numbers of CD8s in the periphery of the ABab-A2 vs ABab-I mice, how were the TCR data normalised?

The y-axes in Figure 4B and 4F were labeled as CD3+CD8+ numbers per μ l blood and CD3+CD8+ numbers per lymphoid organ. The TCR data were normalized in so far as the same number of purified CD8+ T cells per mouse had been sequenced with the help of a mathematical estimator, the iChao1. We now added sentences describing this in detail in the Materials and Methods Section on page 18, lines 462-465.

21) Fig 6 results suggest that there is a large overlap between the CD8+ TCR repertoire of ABab-A2 and ABab-I mice which is extremely surprising given that they have completely different HLA class I, this does not make sense, and it look like the sequencing data is coming from 1 animal only (to confirm) so perhaps some issue with the analysis or result here?

As we had indicated in the legend of Figure 6, there were five mice per group analyzed. Figure 6 show frequency of individual V α and V β genes, both the out-of-frame (pre-selection repertoire) and in-frame (post-selection repertoire). It is apparent that the individual V gene usage is far from random and that the post-selection repertoire mirrors to a large extent the pre-selection repertoire. We were not surprised by these results, because we had observed a similar

bias, when we had sequenced the TCR repertoire of CD4 T cells selected by human HLA-II or mouse MHC-II molecules (Ref. 15).

22) Line 172 “increased smaller” please reword

The sentence on page 8, lines 180-182 has been rephrased to:

“However, the frequency of small size clonotypes had increased in the TCR α (3.5%) and TCR β (4%) repertoire of ABab-A2 compared to ABab-I mice with ...”.

23) Fig 7, is the trend the same if looking at # vs %? The “longer CDR3” for the sub heading refereeing to fig 7 is based on 0.5 nucleotide difference between the 2 mice group, here again if the result is coming from 1 animal the conclusion are not supported.

As we had indicated in the legend of Figure 7, there were five mice per group analyzed.

The differences are statistically highly significant.

On page 18, lines 457-458, we had written: “2-4 μ g of genomic DNA/sample obtained from 3×10^5 mouse (average in mice) CD8+ T cells were deep sequenced.” As the same number of CD8 T cells per group had been sequenced, # vs % would not give additional information.

24) The results would benefit from 1-2 sentences explaining the rationale of what was done followed by the method then results.

In point 18), the reviewer criticizes that there is a lot of rational ... throughout the result which is unusual. Here, the reviewer wishes us to add more rational in the result section. This leaves us in a difficult position what to change, however we feel that we always shortly wrote why and how experiments were done.

25) Do not alternate between HLA-I and MHC-I throughout

We agree and changed accordingly throughout the manuscript.

26) Page 10 “All six HLA I alleles present peptides *ex vivo*” this title is incorrect as allele do not present peptide.

On page 10, line 252, we changed sub-heading to: “All six different HLA-I molecules present peptides *ex vivo*”.

On page 11, line 257 and page 11, line 260 and 261 and in line 265, we deleted the word “model”.

On page 41, line 824, we exchanged “alleles” by “molecules” and deleted the word “model”, also in line 824 and 827.

On page 44, line 838, we deleted the word “model”.

27) Page 11 “T cells for HLA allele recognition bound with its respective model epitope” T cells don't recognize allele, and what is a “model” epitope?

On page 11, lines 259-261, we changed the sentence to: “Co-culture supernatants were analyzed for murine IFN- γ to assess effector T cells for recognition of the respective peptide-HLA molecule.” In line 260, We deleted the word “model”.

In addition:

To avoid ambiguity, we deleted the word “model” connected with antigens or epitopes or TCRs throughout the manuscript.

28) Fig 9, I can't see any no-peptide negative control for any of the cell lines? Were the isolated cells used as APCs for the cell lines? Are the T cell lines HLA-I null? How do you know there is no self-presentation? A control of peptide + transduced T cells (with no ABab cells) is needed to ensure that there is no auto-activation.

The no peptide control was included in Figure 10, where we had restimulated T cells from the peptide-immunized mice with specific or unspecific peptides and observed T cell activation and IFN γ production only with the specific peptides. Therefore, all six TCRs were truly specific for the respective peptide.

Yes, the spleen cells, loaded with the respective peptides, were used as APC for the TCR-transduced T cells. The recipient T cells for TCR transduction were from ABab-A2 mice. They were negative for the six novel HLA-I molecules and self-presentation was excluded.

Collectively Figure 9 and Figure 10 show clearly that CD8 T cells in ABab-I mice can be activated and expanded with effector function in a peptide-specific fashion, where each peptide can be presented by only one of the six HLA-I molecules.

29) In the discussion, it is not clear how these mice will enable epitope-specific TCR discovery any easier than scRNAseq directly from human cells and as such the discussion about TAA requires more information for the readers to understand the link.

On page 3, lines 44-46, we had written: “TAA-specific HLA-I TCRs can preferentially be isolated from transgenic mice, because humans are typically tolerant to TAA and contain usually only low-avidity T cells.” In other words, mice are not tolerant to many human tumor antigens/epitopes, if the sequence of mouse and human differs from each other. Therefore, higher affinity TCRs can be isolated from antigen-negative mice (Ref. 11).

We now additionally modified the discussion on page 12, lines 288-292, by writing: “As the focus has been on HLA-A2 presented peptides for a long time, ABab-I mice may allow to discover novel epitopes as well as respective TCRs, especially for human TAA, which are foreign to the mice. In contrast to mice, which are often negative for human TAA, humans are typically tolerant and bear only low-affinity TCRs.”

30) The discussion is the first time readers are informed that the HLA-I were transferred into the HLA-A2 expressing mice so they have 7x HLA-I molecules but at other times results were referred to as “non-A2”

We disagree here. We had made this clear in the first result paragraph on page 5, new line numbers 90-91 by saying: “The PB ITR-flanked targeting vector carrying six HLA genes was co-injected with the hyperactive PB transposase mRNA into fertilized ABab-A2 mouse oocytes.”

31) It is not clear how the discussion about HLA/TCR evolution contributes much to the story. Indeed, a lot of the discussion reads very different to the intro and how the results story was set up.

We think that we contribute interesting data concerning human T cell biology and thymic selection. To prepare the reader better for the discussion, we added on page 4, lines 74-75 the

following sentence: “ABab-I mice are also useful to investigate human T cell biology and thymic selection under condition of one versus multiple different HLA-I molecules”.

32) Page 12 “ABab-I mice being a useful tool for epitope and TCR discovery” how would the mouse model described here been a better tool for epitope discovery compared to the use of human PBMCs?

As outlined above, there is usually tolerance in humans against TAA, which are self. Very immunogenic epitopes may lead to complete deletional tolerance, precluding the identification of epitopes. Non-tolerant mice may uncover such epitopes.

33) How did the authors compare pre-selection vs post-selection as per the conclusions?

We had written on page 8-9, lines 194-197: V α (Fig. 6A) and V β (Fig. 6B) gene usages were analyzed in both out-of-frame and in-frame TCR- α and - β clonotypes. The former represents the preselection pool, which is an unbiased measure of the frequency by which a given V(D)J recombination event had occurred before selection.

34) The methods would benefit from slightly more details.

We added more details to the Materials and Methods section, which are highlighted in yellow in the manuscript file.

Point-by-point reply to NCOMMS-24-61039B

We would like to thank the Reviewers for their comments and positive notes.

All changes in the main manuscript and supplementary material files are color highlighted in 'yellow'. We want to draw the reviewers' attention to the fact that the following figures have been newly added: Supplementary Figure 2 and Supplementary Figure 8.

The revised manuscript (uploaded as a PDF) has new line numbers due to the addition of some extra sentences. We have addressed all the comments with clear explanations by referencing the updated line and page numbers.

Reviewer #1

We thank Reviewer #1 for the positive feedback and are pleased to hear that he/she is completely satisfied with the updated version of the manuscript.

Reviewer #2

We also appreciate Reviewer #2 for acknowledging that we have appropriately addressed her/his comments.

Reviewer #3

The authors have significantly enhanced the manuscript, but there are still some points to clarify or address with details to help below.

Comments to Reviewer #3:

Comments and questions

1) Now the flow of the story is clearer. The first paragraph of the introduction could be expanded to make it easy to follow for those outside of the field. There are huge jumps, assuming the reader can fill in the knowledge gaps. I.e. "TAA specific HLA-I TCRs can preferentially be isolated from transgenic mice, because humans are typically tolerant to TAA and contain usually only low-avidity T cells". This could be expanded on, to explain that TAA are often self-peptides, and as such during selection high avidity T cells are deleted, leaving only the low avidity ones. As such, these low avidity ones are not optimal in mounting an immune response. To counteract this, this new model ...

On page 3, lines 44-49, we added: "TAA-specific HLA-I TCRs can preferentially be isolated from transgenic mice because humans are typically tolerant to TAA, which are often self-peptides. During thymic selection, high-avidity T cells recognizing self-peptides are deleted to prevent autoimmunity, leaving only low-avidity T cells in circulation. These low-avidity T cells are less effective in mounting a strong immune response against tumors, limiting their therapeutic potential".

2) Line 46 "Transgenic mice are suitable to isolate TCRs from a naïve repertoire through immunization that triggers T cell memory responses" please rephrase, because immunizing will activate T cells (as you say), so you will not be isolating naïve cells (as per the first part of the sentence)...

On page 3, lines 49-51, we had rephrased: “Transgenic mice are suitable for isolating TCRs from a naïve repertoire, which, upon immunization, becomes activated and differentiates into memory T cells, from which TCRs can ultimately be isolated.”

3) Line 60, please add “additional” into the sentence, so as to remind the readers these mice already have A2 expression and now have 6 additional HLA-I molecules.

On page 3, line 64, we had written: “, we here further engineered ABab-A2 mice with six additional human HLA-I alleles by inserting them as one genotype.”

4) Fig 3, why did the authors select B2m rather than staining with the commonly used pan-HLA-I antibody clone W6/32? In addition, the reply to question #15 states “One explanation could be the folding characteristics of full human HLAs vs chimeric molecules, which could hinder epitope accessibility by the anti-human B2m antibody.” This is not correct and was previously shown, also here again w6/32 would have been a better Ab to use. Even if the authors are “are confident that all six HLA-I molecules are functional and expressed” there is no data to demonstrate this presented here.

We now included a new Supplementary Figure 2 staining with pan HLA-ABC (clone: W6/32, Biolegend) and anti-human β 2m antibodies (clone: TÛ99, Beckman Coulter). Because neither the pan HLA-ABC nor the β 2m antibodies allow us to conclude that each of the six HLA-I genes are expressed, we tested HLA expression using both pan HLA-ABC (W6/32) and β 2m antibodies *in vitro* after transiently transfecting murine MCA205 cells with single individual HLA monochains in the same configuration as in the ABab-I mice. These monochain genes had the same configuration and H-2 promoter as the six tandem HLA-I genes. Supplementary Figure 2 shows that all six different HLA-I genes are well expressed in mouse cells. Shown is also IFN- γ treatment of the cells for 48 hours, which led to upregulation of HLA-I expression. On page 6-7, starting lines 136-143, we added: “HLA constructs, designed as single monochains in the same configuration and under the same H-2 promoter as in ABab-I mice...” In Materials and methods section, on page 17, lines 450-454, we added: “HLA surface expression was analyzed *in vitro* in murine MCA205 cells using anti-human pan HLA-ABC (clone: W6/32, Biolegend) and anti-human β 2m antibodies. The β 2m antibody...” We changed the Supplementary file data to reflect new figure numbers.

5) Figure 9, there is still no reference to negative controls (TCR transduced cells + HLA-I cells with NO PEPTIDE) in either the text, figure or figure legend. This needs to be amended. This control (which you have done according to the rebuttal) is critical to show the transduced TCR clones are not non-specifically responding to the HLA-I expressing cells. It would be good to show some raw data (FACS Plots) either in the main figure or in supplemental data.

We appreciate the reviewer's concern regarding the inclusion of negative controls (TCR-transduced cells + HLA-I-expressing cells with NO peptide). While these controls were not included in the final set of experiments shown in Figure 9, they were performed in our initial validation phase for a subset of TCRs (TCRs against 3/7 HLA-I). Specifically, three TCRs from the seven shown in Figure 9 were tested under NO peptide conditions, and no non-specific responses were observed. We have now included these data as a new Supplementary Figure 8, which demonstrates that TCR-transduced T cells do not produce IFN- γ in the absence of cognate peptide, confirming the specificity of the response. We inserted additional sentences in results section accordingly.

On page 11, starting line 270 and lines 272-276, we included: “No IFN- γ production was detected when three of the TCRs was tested in the absence of their cognate peptides, confirming the specificity of TCR recognition (Supplementary Fig. 8). Upon recognition of peptide-bound HLAs, all six TCR-transduced T cells produced IFN- γ in a cognate HLA-peptide-TCR manner (Fig. 9 and Supplementary Fig. 8), proving functional activity.”

On page 19, in Materials and Methods Section, in lines 511-513, we had included: “As negative controls, TCR-untransduced T cells from AB*ab*-A2 mice pulsed with peptides and TCR-transduced T cells co-cultured with HLA-I cells without cognate peptides were used.”

We wish to point out that for each of the six TCR it was shown in the original publication that they recognized cells only if both the respective HLA-I molecule and the peptide was present. Figure 9 and Figure 10 leave no doubt that all six TCR-HLA-peptide combinations are specific. Readout in Figure 9 was ELISA; therefore, we cannot show FACS plots.

We changed the Supplementary file data to reflect new figure numbers.

6) Please start the discussion with a recap of the rationale and why this is being done. Ie we want to generate new tools to be able to identify TCRs that could be used for therapy as human derived ones are low avidity.

It is now much clearer the role of using these murine derived TCRs for ATT as they may be of higher avidity than TCRs found in humans (however this remains an hypothesis), and the addition of the discussion about off target effects is interesting. However, the authors discuss ways they have tried to reduce off target effects (ie alanine scan) but it would be good to broaden this to how these would need to be tested by anyone before being used for therapeutics.

I feel that the flow of the discussion is a bit disjointed. Consider reorganising to enhance readability, a suggested flow below

- Rationale for why you want to make these tools
- How you made these tools
- Why you selected these HLAs and protocols
- Then talk about the TCRs, the caveats around tolerance and testing cross-reactivity (currently this is quite up front, before the discussion on the model itself)
- Finally, how else these mice could be useful.

We thought for a long time how to write the discussion and we think that there are many ways of writing a good discussion. We thought it was important to highlight the human T cell biology findings. The described AB*ab*-I model is not only useful for translational purposes but as well reveal interesting T cell biology findings, as seven different HLA molecules train the TCR repertoire. So, we decided to make only minor changes in the discussion.

On page 12, lines 291-295, to start the discussion with a rationale recap, we added: “The AB*ab*-I transgenic mouse model developed in this study addresses a key challenge in advancing adoptive T cell therapy by overcoming the limitations of human-derived TCRs, which often exhibit low avidity, hindering clinical translation. In contrast, transgenic models like the novel AB*ab*-I mice provide a naïve, non-tolerant TCR repertoire that, upon immunization, can generate high-affinity TCRs suitable for therapeutic use.”

On page 12, lines 310-317, broadly describes how pre-clinical off target TCR assessment not only alanine scan, but alloreactivity and self-peptide screening. So, we added, on page 12, in line 313, we end the sentence with: “prior to clinical translation.”

On page 16, lines 404-417, in Materials and Methods, under study design, we describe how we made the tool, HLA selection and as well the protocol. To avoid redundancy, we want to leave the discussion as it is for now with respect to elaborating the methodology.

In general, we structured the discussion as below:

- We described the key translational purpose (as a recap of rationale) as per request from Reviewer #3.
- Highlighted transposon technology was used to produce these mice since we discuss our breeding strategy later in the discussion.
- Discussed the advantages of ABab-I and the TCRs derived over previously existing models including off-target assessments (our assays make a robust platform for clinical translation).
- Discussed breeding strategy on how we stabilized our mouse model, to maintain a stable CD8 repertoire; ABab-I is a robust tool for TCR isolation and clinical translation.
- Highlighted and discussed in detail the T cell development of human TCRs trained on human HLA haplotype in a mouse closely resembling human situation as well as all the interesting T cell biology findings. As an outlook for T cell biology findings, further development of 'single' HLA-bearing ABab-I-'X' mice, we can investigate, if any, the HLA-I/TCR-V β intrinsic affinity bias.

We decided on the above structure based on the comments, critiques, and questions from Reviewers #1 and #2, who were satisfied with the discussion.

Therefore, we are comfortable with the current organization of the discussion.

Point-by-point reply to NCOMMS-24-61039C

Changes in the main manuscript and supplementary material files are color highlighted in 'yellow'.

Reviewer #3

The authors have significantly enhanced the manuscript, but there are still some points to clarify or address with details to help below.

1) Now the flow of the story is clearer. The first paragraph of the introduction could be expanded to make it easy to follow for those outside of the field. There are huge jumps, assuming the reader can fill in the knowledge gaps. I.e. "TAA specific HLA-I TCRs can preferentially be isolated from transgenic mice, because humans are typically tolerant to TAA and contain usually only low-avidity T cells". This could be expanded on, to explain that TAA are often self-peptides, and as such during selection high avidity T cells are deleted, leaving only the low avidity ones. As such, these low avidity ones are not optimal in mounting an immune response. To counteract this, this new model ...

On page 3, lines 44-49, we added: "TAA-specific HLA-I TCRs can preferentially be isolated from transgenic mice because humans are typically tolerant to TAA, which are often self-peptides. During thymic selection, high-avidity T cells recognizing self-peptides are deleted to prevent autoimmunity, leaving only low-avidity T cells in circulation. These low-avidity T cells are less effective in mounting a strong immune response against tumors, limiting their therapeutic potential".

2) Line 46 "Transgenic mice are suitable to isolate TCRs from a naïve repertoire through immunization that triggers T cell memory responses" please rephrase, because immunizing will activate T cells (as you say), so you will not be isolating naïve cells (as per the first part of the sentence) ...

On page 3, lines 49-51, we had rephrased: "Transgenic mice are suitable for isolating TCRs from a naïve repertoire, which, upon immunization, becomes activated and differentiates into memory T cells, from which TCRs can ultimately be isolated."

3) Line 60, please add "additional" into the sentence, so as to remind the readers these mice already have A2 expression and now have 6 additional HLA-I molecules.

On page 3, line 64, we had written: "we here further engineered ABab-A2 mice with six additional human HLA-I alleles by inserting them as one genotype."

4) Fig 3, why did the authors select B2m rather than staining with the commonly used pan-HLA-I antibody clone W6/32? In addition, the reply to question #15 states "One explanation could be the folding characteristics of full human HLAs vs chimeric molecules, which could hinder epitope accessibility by the anti-human B2m antibody." This is not correct and was previously shown, also here again w6/32 would have been a better Ab to use. Even if the authors are "are confident that all six HLA-I molecules are functional and expressed" there is no data to demonstrate this presented here.

The W6/32 antibody stains all HLA-I molecules indirectly by targeting residues in the N-terminus of the human β 2-microglobulin (β 2m) molecule. As a result, it does not allow for the precise identification of which HLA-I molecules are expressed on the cell surface (Reference: Shields MJ, Ribaldo RK. Tissue Antigens, 1998. The article is attached as 'Auxiliary Reference File' (W6/32 Ab paper) at the end).

Therefore, whether one uses the β 2m antibody or the W6/32 antibody to stain HLA-I makes little difference, as both methods are indirect and non-specific for any of the six HLA-I molecules. Additionally, the referenced article, which maps the epitope for W6/32, demonstrates that W6/32 does not react with recombinant human β 2m—which is relevant in our mouse model.

Figure showing the lack of reactivity of the W6/32 antibody when recombinant human β 2m was used, as published in the referenced paper cited above.

This could explain why W6/32 did not stain well in our experiments. Consequently, we used the β 2m antibody as a surrogate, arguing that both stain for the same β 2m N'-terminus, as shown in Supplementary Figure 2B. We added these data in the new Supplementary Figure 2. We now included a new Supplementary Figure 2 showing pan HLA-ABC (clone: W6/32, Biolegend) and anti-human β 2m antibodies (clone: TÛ99, Beckman Coulter) staining. Because neither the pan HLA-ABC nor the β 2m antibodies does allow to conclude that each of the six MHC I genes are expressed, we tested HLA expression using both pan HLA-ABC (W6/32) and β 2m antibodies *in vitro* after transiently transfecting murine MCA205 cells with single individual HLA monochains in the same configuration as in the *ABab-I* mice. These monochain genes had the same configuration and H2 promoter as the six tandem HLA I genes. Supplementary Figure 2 shows that all six different HLA-I genes are well expressed in mouse cells. Shown is also IFN- γ treatment of the cells for 48 hours, which led to upregulation of HLA-I expression.

On page 6-7, lines 136-143. We changed the Supplementary file data to reflect new figure numbers.

On page 17, in Materials and methods section, lines 452-456, we added: “HLA surface expression was analyzed *in vitro* in murine MCA205 cells using anti-human pan HLA-ABC (clone: W6/32, Biolegend) and anti-human β 2m antibodies (clone: TÛ99, Beckman Coulter). The β 2m antibody...”

5) Figure 9, there is still no reference to negative controls (TCR transduced cells + HLA-I cells with NO PEPTIDE) in either the text, figure or figure legend. This needs to be amended. This control (which you have done according to the rebuttal) is critical to show the transduced TCR clones are not non-specifically responding to the HLA-I expressing cells. It would be good to show some raw data (FACS Plots) either in the main figure or in supplemental data.

We appreciate the reviewer's concern regarding the inclusion of negative controls (TCR-transduced cells + HLA-I-expressing cells with NO peptide). While these controls were not included in the final set of experiments shown in Figure 9, they were performed in our initial validation phase for a subset of TCRs (TCRs against 3/7 HLA-I's). Specifically, three TCRs from the seven shown in Figure 9 were tested under NO peptide conditions, and no non-specific responses were observed.

We have now included these data as a new Supplementary Figure 8, which demonstrates that TCR-transduced T cells do not produce IFN- γ in the absence of cognate peptide, confirming the specificity of the response. We inserted additional sentences in results section accordingly. On page 11, starting lines 272-276, we included: "No IFN- γ production was detected when three of the TCRs was tested in the absence of their cognate peptides, confirming the specificity of TCR recognition (Supplementary Fig. 8). Upon recognition of peptide-bound HLAs, all six TCR-transduced T cells produced IFN- γ in a cognate HLA-peptide-TCR manner (Fig. 9 and Supplementary Fig. 8), proving functional activity."

On page 19, in Materials and Methods Section, in lines 513-515, we had included: "As negative controls, TCR-untransduced T cells from ABab-A2 mice and TCR-transduced cells co-cultured with HLA-I cells without cognate peptides were used." We changed the Supplementary file data to reflect new figure numbers.

We wish to point out that for each of the TCRs it was shown in the original publication that they recognized cells only if both the respective HLA I molecule and the peptide was present. Below are the no peptide control data for the remaining TCRs, as published in the original publications.

No peptide control data. Referring to Figure 9 and Suppl. Figure 8

Model TCRs restricted towards epitopes presented by HLA alleles in ABab-I mice					W'out peptide control
HLA restriction	TCR chains	V(D)J recombined regions	CDR3 regions	Epitope sequences (IC ₅₀)	
A*03:01	TCR Va chain TCR V β chain	TRAV9-2-TRAJ30-1 TRBV13-1-TRBD2-1-TRBJ2-3	CALSDRERDDKIIF CASSHEGLAGEFF	KMRMRMRMR (29.18 nM) mCALRp7 neoantigen	Original publication below
A*11:01	TCR Va chain TCR V β chain	TRAV3-3-TRAJ17-1 TRBV4-1-TRBD2-1-TRBJ2-1	CAVSGGTNSAGNKLTIF CASSRDWGPAEQFF	VVGAVGVGK (65.47 nM) KRAS G12V neoantigen	Current manuscript
B*07:02	TCR Va chain TCR V β chain	TRAV17-1-TRAJ12-1 TRBV7-9-TRBD1-1-TRBJ2-1	CATVIRMDSSYKLIIF CASSLIGVSSYNEQFF	TPRVTGGGAM (3.86 nM) CMV pp65 viral antigen	Original publication below
B*15:01	TCR Va chain TCR V β chain	TRAV17-1-TRAJ47-2 TRBV6-5-TRBD1-1-TRBJ2-5	CAESEYGKLVF CASSYRQQTQYF	SAFRCFIVY (118.20 nM) HPV16 E5 viral antigen	Original publication below
C*04:01	TCR Va chain TCR V β chain	TRAV26-2-TRAJ49-1 TRBV15-2-TRBD2-2-TRBJ2-1	CILRDNTGNQFYF CATSRDGSSYNEQFF	QYDPVAALF (499.34 nM) CMV pp65 viral antigen	Original publication below
C*07:02	TCR Va chain TCR V β chain	TRAV13-1-TRAJ34-1 TRBV25-1-TRBD2-1-TRBJ2-7	CAASGATDKLIIF CASSGGHEQYF	VRIHGLYLIL (647.00 nM) MAGE-A12 shared antigen	Current manuscript
A*02:01	TCR Va chain TCR V β chain	TRAV5-TRAJ41 TRBV28-TRBJ2-7	CAESIGSNSGYALNF CASRGLAGYEQYF	KVLEYVIKV (6.91 nM) MAGE-A1 shared antigen	Current manuscript

Table showing in which 3/7 TCR co-cultures 'no peptide' negative controls were included in this manuscript, and in which 4/7 TCR cases this control has already been published.

No peptide control data. HLA-A*03:01 restricted TCR co-culture

successfully identified thirteen HLA-B*07:02-restricted CALR-RMR and two HLA-A*03:01-restricted CALRp7 re-active T cell clones. We proceeded to isolate the TCRαβ variable gene sequences from the highest IFN-γ-producing T cell clones (CALR-RMR clone 4, CALR-RMR clone 20, CALRp7 clone 2 and CALRp7 clone 3) using TCR gene capture technology (Additional file 6).

Citation:

Tubb *et al. Journal for ImmunoTherapy of Cancer* (2018) 6:70
<https://doi.org/10.1186/s40425-018-0386-y>

Figure showing data from A*03:01-mCALR-restricted TCRs, with irrelevant peptides as 'no peptide' negative controls, as presented in the *Tubb et al.* paper.

No peptide control data. HLA-B*07:02 & B*15:01 restricted TCRs co-culture

Figure 3. Re-expression of TCR chains in peripheral blood lymphocytes and analysis of TCR function. (A) Map of TRA and TRB chain genes molecularly cloned into the γ-retrovirus vector MP71. Human TRAV and TRBV gene segments were fused to murine TRAC and TRBC gene sequences. Flow cytometry plots of primary T cells transduced with different TRA/TRB combinations found in (B) E5-B*15:01- and (C) pp65-B*07:02-specific T cells are shown. Numbers in the upper right corner indicate percentages of expression of murine TRBC chains in CD8⁺ T cells. T cells transduced with different TRA/TRB combinations were co-cultured for 18 h with (D) K562-B*15:01 target cells expressing HPV16 E5 or no antigen and with (E) K562-B*07:02 target cells expressing CMV pp65 or no antigen. Shown is IFN-γ release of T cells measured by ELISA.

Citation:

Unbiased Identification of T-Cell Receptors Targeting Immunodominant Peptide-MHC Complexes for T-Cell Receptor Immunotherapy

Felix K.M. Lorenz¹, Christian Ellinger², Elisa Kieback³, Susanne Wilde^{2,1}, Maria Lietz², Dolores J. Schendel², and Wolfgang Uckert^{1,4,5,6}

¹Max-DeBock Center for Molecular Medicine in the Neurology Association, Berlin, Germany; ²Institute for Molecular Immunology, Helmholtz-Zentrum Munich, Munich, Germany; ³Institute of Biologie, Humboldt-University Berlin, Berlin, Germany; ⁴State Institute of Health, Berlin, Germany; ⁵Present address: Medigene Immunotherapies GmbH, Planegg/Martinsried, Germany;

Figure showing data from B07:02-CMVpp65-restricted and B15:01-HPVE5-restricted TCRs, with 'no peptide' negative controls, as presented in the *Lorenz et al.* paper.

No peptide control data. HLA-C*04:01 restricted TCR co-culture

Figure 5. Cytolytic activity of the CTL lines generated by stimulation with CD40-B cells pulsed with newly identified epitopes. CD8⁺ T cells from CMV-seropositive donors were stimulated with peptide-pulsed autologous CD40-B cells. CMVpp65-derived synthetic peptides used were CEDVPSGKL (pp65²³²⁻²⁴⁰) to be presented by HLA-B*4001 (A), HERNGFTVL (pp65²⁶⁷⁻²⁷⁵) by HLA-B*4001 (B), AELEGVWQPA (pp65⁵²⁵⁻⁵³⁴) by HLA-B*4006 (C), QYDPVAALF (pp65³⁴¹⁻³⁴⁹) by HLA-Cw*0401 (D), VVCAHELVC (pp65¹⁹⁸⁻²⁰⁶) by HLA-Cw*0801 (E) or Cw*1502 (F). One week after the third stimulation, cytotoxicity of CTL lines was assessed by ⁵¹Cr release assay against LCL/pp65 (■), LCL/EGFP (●), or 1 μM peptide-pulsed LCL/EGFP (○) over a range of E/T ratios as indicated. The origin of target cells was autologous (panel F) or allogeneic [panels A-B, N06 (B*4001+); panel D, P01(Cw*0401+); panels C and E, N03(B*4006, Cw*0801+)].

Citation:

IMMUNOBIOLOGY

Identification of novel CTL epitopes of CMV-pp65 presented by a variety of HLA alleles

Etsi Kondo, Yoshiaki Akatsuka, Kiyotaka Kuzushima, Kurio Tsujimura, Ehoji Asakura, Kohji Tajima, Yoshitoyo Kagami, Yoshihisa Kodera, Mitsuru Tamoto, Yasuo Morinaka, and Toshitada Takahashi

Figure showing data from C*04:01-CMVpp65-restricted TCR, with no peptide negative controls, as presented in the *Kondo et al.* paper.

Additionally, on page 19, in Materials and Methods Section, in lines 515-516, we had included: “In some cases, we included no peptide as negative controls, and in other cases, this control had already been published.” We cited the respective articles in line 516.

Figure 9 and Figure 10 leave no doubt that all six TCR-HLA-peptide combinations are specific. Readout in Figure 9 was ELISA; therefore, we cannot show FACS plots.

We would like to point out that new no-peptide control experiments are *in vivo* mouse experiments and have to be approved by authorities. This is in our country a major act. Writing an animal application takes at least 2-3 months. Reviewing by authorities requires another 3 months at least but can be much longer. We needed to say that the results of the experiments are not known yet, otherwise such an animal application would unlikely be approved, because the experiments would be judged as unethically. In any case, it would delay publication of our manuscript 6-9 months.

6) Please start the discussion with a recap of the rationale and why this is being done. If we want to generate new tools to be able to identify TCRs that could be used for therapy as human derived ones are low avidity.

It is now much clearer the role of using these murine derived TCRs for ATT as they may be of higher avidity than TCRs found in humans (however this remains an hypothesis), and the addition of the discussion about off target effects is interesting. However, the authors discuss ways they have tried to reduce off target effects (ie alanine scan) but it would be good to broaden this to how these would need to be tested by anyone before being used for therapeutics.

I feel that the flow of the discussion is a bit disjointed. Consider reorganising to enhance readability, a suggested flow below

- Rationale for why you want to make these tools

- How you made these tools
- Why you selected these HLAs and protocols
- Then talk about the TCRs, the caveats around tolerance and testing cross-reactivity (currently this is quite up front, before the discussion on the model itself)
- Finally, how else these mice could be useful.

Discussion has been changed accordingly.

On page 12, lines 291-295, we had recap of the rationale by adding sentences: “The ABab-I transgenic mouse model developed in this study addresses a key challenge in advancing adoptive T cell therapy by overcoming the limitations of human-derived TCRs, which often exhibit low avidity, hindering clinical translation. In contrast, transgenic models like the novel ABab-I mice provide a naïve, non-tolerant TCR repertoire that, upon immunization, can generate high-affinity TCRs suitable for therapeutic use”.

Originally from page 13, lines 323-334 has been reorganized to page 12 as lines 297-308. Following, page 12, lines 308-310, we added: “The six HLA class I genes HLA-A*03:01, -A*11:01, -B*07:02, -B*15:01, -C*04:01, and -C*07:02 were chosen due to their high frequency among the world population.

Originally from page 12, lines 297-318 has been reorganized as lines 311-331 (page 12-13). Here, we discuss tolerance, testing off-target reactivity and toxicity prior to clinical translation. We finally describe how these mice could be useful from page 13-15, lines 337-400.

Reference: Shields MJ, Ribaud RK. Tissue Antigens, 1998.

[REDACTED]

Point-by-point reply to Reviewer #3's comments_NCOMMS-24-61039D

Reviewer #3

We thank Reviewer #3 for the positive feedback and are pleased to hear that he/she is completely satisfied with our answers updated in the current version of the manuscript.

Reviewer #3 (Remarks to the Author):

The authors have answered my questions, provided a detailed rebuttal and have made their manuscript clearer. I have no further questions, just some corrections to incorrect comments regarding W6/32 Ab use and binding from the rebuttal.

Minor comments (about the rebuttal only)

1. Line 138 – The anti-human pan HLA-ABC, is it pan HLA class I, even for non-classical HLA-I or only for classical HLA-I? W6/32 is a pan HLA class I molecules, not only ABC.

We agree with the Reviewer that the pan HLA-ABC antibody could stain non-classical HLA-I due to its targeting epitope located in the N-terminus of the human β 2-microglobulin (β 2m) molecule. In principle, all non-classical HLA-I molecules associated with β 2m can be stained using the W6/32 antibody.

However, in Line 138, to align with the standard nomenclature for the W6/32 clone—'anti-human pan HLA-ABC' as provided by the antibody supplier (in this case, BioLegend)—and to help readers find the correct antibody for their assays, we will retain our wording in Line 138 as it currently reads: “expression using anti-human pan HLA-ABC and β 2m antibodies. These data showed that each of...”, which describes the W6/32 antibody best.

We have depicted this in the new Supplementary Figure 2. We transfected the murine MCA205 cell line with single HLA-I monochains and stained for classical HLA-I using W6/32. Classical HLA-I expression is described through this supplementary data, and therefore, we feel it is not necessary to mention non-classical HLA-I molecules in this context. Of note, there are no non-classical HLA-I molecules present in the mice.

2. “The W6/32 antibody stains all HLA-I molecules indirectly by targeting residues in the N-terminus of the human β 2-microglobulin (β 2m) molecule.”

This is incorrect, the W6/32 Ab bind the HLA-I molecule itself, and the β 2m, making it specific to HLA class I, both classical and non-classical, but not to HLA-like molecule such as CD1 or MR1 molecules. The crystal structure of the W6/32-HLA-I complex is showing this with great details (PDB code 7TOL).

We thank the Reviewer for sharing the recent and intriguing crystal structure data of the W6/32-HLA-I complex. However, as mentioned in our previous rebuttal, this information does not directly aid in uniquely differentiating the expression of all six HLA-I molecules through staining. This is the primary reason we employed the transfection method, introducing the six HLA-I molecules as single monochains each time, to present the in vitro staining data shown in Supplementary Figure 2. To further our understanding, we will explore this crystal structure data in more depth, as it remains unclear to us exactly how and which residues from the light chain of the W6/32 antibody interact with which specific residues within the 276 described amino acids of the HLA-I antigen.

3. “As a result, it does not allow for the precise identification of which HLA-I molecules are expressed on the cell surface”

This is the case for all pan-Ab, while it does not allow for the precise identification of which HLA-I is expressed, it does show that there is an HLA-I molecule expressed.

We agree with the Reviewer’s comment. However, we would like to emphasize, as mentioned in our previous rebuttal, that the β 2m antibody, although not uniquely, also demonstrates that an HLA-I molecule is expressed. Since β 2m is directly fused to the HLA-I heavy chain domains in our ABab-I mice, we can clearly show HLA-I expression, supported by functional data.

4. “Therefore, whether one uses the β 2m antibody or the W6/32 antibody to stain HLA-I makes little difference, as both methods are indirect and non-specific for any of the six HLA-I molecules.”

β 2m can be associated to MHC-like molecule such as CD1 or MR1, so staining only with β 2m Ab is a real issue, especially for this study.

We respectfully disagree with the Reviewer’s comment: “staining only with β 2m Ab is a real issue, especially for this study.”

We are aware that in mice, non-classical molecules such as murine CD1 are associated with murine β 2m and are encoded by the mCD1.1 and mCD1.2 genes (Brutkiewicz RR *et al.*, J. Exp. Med., 1995). However, we would like to highlight the following critical points to clarify why β 2m antibody staining is not a concern in our study:

1. In the ABab-I mice described here, each of the six HLA-I molecules is expressed as a fusion protein with human β 2m. The antibody clone TÛ99, used in our experiments (Figure 3 and Supplementary Figure 2), is specific to human β 2m and does not cross-react with β 2m from other species. We clearly show HLA-I expression.

2. The ABab-I mice were generated on an ABab-A2 background, which includes a double knockout for murine MHC-I and murine β 2m (Li LP *et al.*, Nat. Med., 2010). As murine CD1 molecules require murine β 2m for surface expression, and no free human β 2m is available due to its fusion to HLA-I, off-target staining is highly unlikely.

Given these conditions, there is no possibility that the TÛ99 anti-human β 2m antibody stains murine CD1.1, CD1.2, or MR1 in our model.

5. “Additionally, the referenced article, which maps the epitope for W6/32, demonstrates that W6/32 does not react with recombinant human β 2m” This is incorrect. All the work using recombinant human β 2m show that the w6/32, used for example in Surface plasmon resonance assay” bind the human β 2m. The structure of W6/32 with HLA-I (PDB code 7T0L) is in complex with recombinant human β 2m, clearly showing binding.

As mentioned above in Response No. 2, we plan to explore the crystal structure data in greater depth, as it remains unclear to us exactly how and which residues from the light chain of the W6/32 antibody interact with which specific residues within the 276 amino acids described for the HLA-I antigen.

We thank the Reviewer for pointing out the relevance of Surface Plasmon Resonance (SPR) assays. In our previous rebuttal, we stated that “W6/32 did not stain well in our experiments,” but we did not claim that no staining was observed. This comment was intended to support our observation that, in our hands, the human β 2m antibody produced stronger staining (higher MFI, Supplementary Figure 2).

In support of our flow cytometry findings, van Osch TLJ *et al.* demonstrated using SPR that the most stabilizing interactions were not observed with the W6/32 antibody (van Osch TLJ *et al.*, Haematologica, 2022).

Figure 2 from the above referenced article shows: 'Surface plasmon resonance visualizes the co-binding and stabilization of multiple anti-HLA monoclonal antibodies to HLA-A*02:01'.

6. “This could explain why W6/32 did not stain well in our experiments.”

This comment is very confusing, as this is not what Supp Fig 2 shows, the w6/32 staining is comparable to the b2m-Ab staining.

As mentioned above in Response No. 5 and in our previous rebuttal, we stated that “W6/32 did not stain well in our experiments,” but we did not claim that no staining was observed.

Hence, we respectfully disagree and would like to highlight that the human β 2m antibody produced significantly stronger staining, with higher mean fluorescence intensity (MFI), as shown in Supplementary Figure 2. This comment was intended to support our observation based on results obtained in our hands.